

# Asymptotic T-duality in three dimensions

**Stéphane Detournay[1]★, José Figueroa[1,2]† and Alejandro Vilar López[3]‡**

**1** Physique Théorique et Mathématique and International Solvay Institutes,
Université Libre de Bruxelles (ULB), C.P. 231, 1050 Brussels, Belgium
**2** Departamento de Física, Universidad de Concepción, Casilla 160-C, Concepción, Chile
**3** Department of Physics and Astronomy, University of British Columbia,
6224 Agricultural Road, Vancouver, B.C. V6T 1Z1, Canada

★ sdetourn@ulb.ac.be , † jose.figueroa.silva@ulb.be , ‡ alejandro.vilarlopez@ubc.ca

## Abstract

In (super)gravity theories, T-duality relates solutions with an exact isometry which can have wildly different asymptotic behaviors: A well-known example is the duality between BTZ black holes and (non-extremal) three-dimensional black strings. Using this dual pair, we show how the knowledge of a phase space which includes one set of solutions (here, BTZ black holes embedded in the Brown-Henneaux phase space) allows to obtain a phase space for the dual set via an asymptotic notion of T-duality. The resulting asymptotic symmetry algebras can be very different. For our particular example, we find a large algebra of symmetries for the black string phase space which includes as subalgebras $\mathfrak{bms}_2$, $\mathfrak{bms}_3$, and a twisted warped conformal algebra. On the way, we show that a chiral half of the Brown-Henneaux boundary conditions are dual to the Compère-Song-Strominger ones.



# 1  Introduction

The identification of simple, lower-dimensional gravitational models which capture the essence of physically relevant higher-dimensional spacetimes has been a very fruitful enterprise. Three dimensional theories are a sweet spot for this approach, and over the years they have illuminated questions in a wide range of topics: The asymptotic symmetry structure of AdS gravity in the pre-holographic era [1], black hole microstate countings using the symmetries of the underlying dual [2–5], linear-response theory in black hole backgrounds [6], string theory embeddings [7–11], or the relation between asymptotic and worldsheet symmetries [12–16], to name a few. An early three-dimensional toy model, inaugurating the arXiv and even predating the BTZ black hole [17], was the Horne-Horowitz black string [18]. It is a $(2 + 1)$-dimensional solution to the string theory low-energy effective action which shares many features with higher-dimensional Reissner-Nordström black holes: It has a non-trivial causal structure with outer and inner horizons, a timelike curvature singularity, thermal behavior, and vanishing curvature at large radius (making it, in a certain sense, asymptotically flat).

Despite all of this, or perhaps because of it, the three-dimensional black string has been less studied than its asymptotically AdS$_3$ black hole counterpart: The BTZ black hole, which has played a pivotal role in many developments of AdS/CFT [2, 6, 19, 20]. Still, the three-dimensional black string enjoys several remarkable properties and a relatively close relationship with BTZ black holes. It was introduced as an exact string theory background, via the construction of a gauged version of the $SL(2, \mathbb{R}) \times \mathbb{R}$ Wess-Zumino-Witten CFT [18]. It was later recognized that black strings could also be obtained as the target space of marginal deformations of the $SL(2, \mathbb{R})$ WZW worldsheet theory describing BTZ black holes [21]. Another relation came from the observation that a general class of black strings results from TsT transformations [22, 23] applied to BTZ black holes, and this resulted in the proposal that TsT-transformed AdS$_3$ backgrounds are holographically dual to single-trace $T\bar{T}$ deformations of the boundary CFT$_2$ [24, 25] (see also [26]).

In this paper, we will exploit a different connection between three-dimensional black strings and BTZ black holes [7]: They are related by the low-energy manifestation of T-duality implemented by Buscher rules [27, 28]. These transformations map any solution of the low energy string equations with a translational Killing vector to another solution. Under certain conditions, the related backgrounds actually correspond to equivalent worldsheet theories [29], but we would like to emphasize from the start that in this paper we will not be dealing with the underlying BTZ CFT [8, 16]. We will instead restrict ourselves to the (super)gravity realm in which Buscher rules provide a map generating new solutions from existing ones (with an isometry). Previous works used this perspective to discuss invariance of the thermodynamic properties of horizons under T-duality [30], even in the presence of $\alpha'$ corrections [31, 32]. We will now exploit a different property of the duality which makes it particularly interesting for the study of asymptotic structures: It can relate solutions with wildly different asymptotic behavior.

At a classical level, asymptotic boundary conditions for gravitational theories determine the symmetries present in the phase space [33–35]. These are in turn an essential hint towards a

quantum formulation of the theory, as the well-known example of Brown-Henneaux boundary conditions in AdS$_3$ illustrates [1]: The asymptotic symmetry algebra should be realized in any putative dual quantum mechanical description. Given that BTZ black holes belong to the well-understood Brown-Henneaux phase space, and that they can be related via T-duality to the Horne-Horowitz black strings, our goal will be to use T-duality as a tool to obtain a phase space for the black strings. This is a novel approach, different to those used to define boundary conditions for the three-dimensional black strings in the past [36, 37]. The result of our method will be a phase space carrying a new asymptotic symmetry algebra (66)-(67), significantly larger than the ones previously discussed in the literature and including as subsets some well-known subalgebras: $\mathfrak{bms}_2$, $\mathfrak{bms}_3$ and a twisted warped conformal algebra. Notice that, since the black strings are asymptotically flat, this is a symmetry algebra for the three-dimensional low-energy string EFT with asymptotically flat boundary conditions (although, admittedly, the notion of asymptotic flatness is slightly non-standard due to the Killing fields having diverging norm at infinity; see section 4 for the details).

It is important to emphasize that this symmetry algebra is different from the one of the original Brown-Henneaux phase space, which as it is well-known carries two copies of the Virasoro algebra. This may seem puzzling at first, particularly if one thinks of T-duality in the stringy sense as relating equivalent backgrounds. However, we are here just applying Buscher rules to the low-energy (super)gravity fields, and we are doing so in a restricted and asymptotic sense (to be explained in section 4) which allows to generate new boundary conditions from existing ones. As a consequence, solutions in the original Brown-Henneaux phase space are not one-to-one mapped to solutions in the black string phase space (it is only solutions having an exact isometry that get mapped between both sides). It is in this sense that we dub our procedure *asymptotic T-duality*. We believe similar ideas can be applicable to other sets of dual pairs (not necessarily in three dimensions), thus making asymptotic T-duality an interesting way to generate new boundary conditions from well-understood ones.

The remainder of this paper is structured as follows. In section 2, we uplift the standard Brown-Henneaux analysis of asymptotically AdS$_3$ boundary conditions to the universal bosonic sector of the low energy effective field theory of strings – that is, we include a dilaton and a Kalb-Ramond two-form. Despite obtaining the same set of charges as in the original work, forming two Virasoro towers, the section sets up the notation and conventions that we will use later when applying Buscher rules. In section 3, as a warm-up, we consider a situation in which we have a complete phase space with an exact Killing vector: A chiral subset of the previous AdS$_3$ solution space. Upon dualizing in the direction of the exact Killing vector, we obtain a dual phase space which turns out to be exactly the one obtained by Compère, Song and Strominger (CSS) in [38]. Section 4 constitutes the bulk of this work, and it shows by means of an example how T-duality can be used to generate new boundary conditions from existing ones. From the full set of Brown-Henneaux boundary conditions, and dualizing along an angular direction which provides an asymptotic isometry, we obtain a new set of boundary conditions for the low energy string effective field theory. The non-extremal black strings, being T-dual to BTZ black holes, are included within the phase space defined by these boundary conditions. The asymptotic symmetry algebra has four infinite towers of charges, which combine in a way to include $\mathfrak{bms}_2$, $\mathfrak{bms}_3$, and a twisted version of the warped conformal algebra as subalgebras. We conclude with a discussion in section 5. Three appendices provide extra details relevant at different points of the paper. Appendix A shows that, in a phase space with an exact Killing vector, asymptotic symmetry transformations are the same before and after applying a T-duality transformation. Appendix B discusses in detail how black strings fit within the boundary conditions of section 4. Finally, appendix C collects results needed to evaluate the algebra of the charges in section 4.

## 2 Brown-Henneaux boundary conditions for the string EFT

In this section, we will generalize the classical analysis of asymptotically $AdS_3$ boundary conditions by Brown and Henneaux [1] so that it applies to the low energy effective theory governing the universal massless NS-NS sector of string theory. This problem has already been considered in previous works [39], and our results will be compatible with them. The role of this section is mainly to set the stage for a later application of T-duality transformations to the Brown-Henneaux boundary conditions, so in particular the notation and methods presented here will recurrently appear throughout the paper.

Consider the low energy string effective theory governing the NS-NS sector,

$$L = \frac{1}{2\kappa_N^2} \mathcal{L}_0 \, \epsilon \,, \qquad \mathcal{L}_0 = R - 2\Lambda_0 e^{4\Phi} - 4(\partial \Phi)^2 - \frac{1}{12} e^{-8\Phi} H^2 \,. \tag{1}$$

The basic fields are the dilaton $\Phi$, the metric $G_{MN}$ and the Kalb-Ramond two-form $B_{MN}$ – which appears through its field strength $H = dB$ in the term $H^2 = H^{MNR} H_{MNR}$. We have also denoted by $\epsilon$ the volume form of the manifold, and $\kappa_N^2$ is the gravitational coupling. Note that we always work in Einstein frame. Varying the fields we obtain

$$\delta L = \frac{1}{2\kappa_N^2} \left[ -\delta G_{MN} \mathcal{E}_G^{MN} + 2\delta \Phi \, \mathcal{E}_\Phi + \frac{1}{2} \delta B_{PQ} \mathcal{E}_B^{PQ} \right] \epsilon + d\Theta \,, \tag{2}$$

with

$$\mathcal{E}_\Phi = 4\nabla^2 \Phi - 2e^{-8\Phi} h^2 - 4\Lambda_0 e^{4\Phi} \,, \tag{3a}$$

$$\mathcal{E}_B^{PQ} = \epsilon^{PQM} \nabla_M \left( e^{-8\Phi} h \right) \,, \tag{3b}$$

$$\mathcal{E}_G^{MN} = R^{MN} - \frac{1}{2} G^{MN} \mathcal{L}_0 - 4\nabla^M \Phi \nabla^N \Phi + \frac{1}{2} e^{-8\Phi} h^2 G^{MN} \,, \tag{3c}$$

and the boundary contribution is $\Theta = \theta \cdot \epsilon$ with

$$\theta^R = \frac{1}{2\kappa_N^2} \left[ 2G^{L[R} \nabla^{S]} \delta G_{SL} - 8\delta \Phi \nabla^R \Phi - \frac{1}{2} e^{-8\Phi} H^{RPQ} \delta B_{PQ} \right] \,. \tag{4}$$

We are using the notation for differential form calculus of [40], and we have often simplified expressions by taking the scalar dual of the 3-form $H$, $h = -\star H$ (so that $H = h\,\epsilon$).

We impose now boundary conditions inspired by those of [1], in a form adapted to connect with Bañados metrics [41, 42]:[1]

$$ds^2 = \frac{\ell^2}{r^2} dr^2 + r^2 \left( \eta_{ab} + \frac{\ell^2}{r^2} Y_{ab} + \dots \right) dx^a \, dx^b \,, \tag{5a}$$

$$B = \left( \frac{r^2}{C_0} + b(x^a) + \frac{\beta(x^a)}{r^2} + \dots \right) dx^+ \wedge dx^- \,, \tag{5b}$$

$$\Phi = \frac{1}{4} \log \left( \frac{2}{C_0} \right) + \frac{\tilde{Y}(x^a)}{r^2} + \frac{\phi(x^a)}{r^4} + \dots \,, \tag{5c}$$

where $a, b, \dots$ label coordinates in the two-dimensional space orthogonal to the radial direction, and $\eta_{ab}$ is the Minkowski metric in that two-dimensional space. These boundary conditions are defined by two fixed parameters, a length scale $\ell$ setting the asymptotic $AdS_3$

---

[1]These could be more appropriately called *falloffs* for the fields. Only the leading terms are taken to be fixed, subleading terms such as $Y_{ab}$, $b$, $\beta$, $\tilde{Y}$ and $\phi$ are dynamical phase space variables.

radius and a dimensionless constant $C_0$.[2] This sets the charge of the Kalb-Ramond two-form field, $B$, as we will later discuss in more detail. An asymptotic analysis of the equations of motion (3) shows that, in order to have solutions, we must relate the parameters $\ell$ and $C_0$ to the cosmological constant $\Lambda_0$ as

$$\Lambda_0 = -\frac{C_0}{\ell^2}\,. \tag{6}$$

This is analogous to the usual $\Lambda_0 = -1/\ell^2$ value in Einstein gravity, but here modified by the $B$-field charge.[3] Note that we cannot set $C_0 = 0$, since the constant piece in the dilaton would blow up. From now on we will write our expressions in terms of $C_0$ and $\ell$. We have also chosen to use null coordinates for the constant radius surfaces, but they can be traded for more standard ones via $x^\pm = t/\ell \pm \phi$. The choice of Fefferman-Graham gauge, setting $G_{rr} = \ell^2/r^2$ and $G_{ra} = 0$, is convenient but not essential.

The previous set of boundary conditions can be shown to define a well-posed variational problem if we add to our action the following boundary term

$$\mathcal{B} = \frac{1}{\kappa_N^2}\left(K - \frac{1}{\ell}\right)\epsilon_{\partial\mathcal{M}} + \frac{1}{2\kappa_N^2}\left(e^{-8\Phi} \star H\right)B\,, \tag{7}$$

so that when varying we consider the full action

$$S = \int_{\mathcal{M}} L + \int_{\partial\mathcal{M}} \mathcal{B}\,. \tag{8}$$

Indeed, a first order variation produces a boundary term of the form

$$
\begin{aligned}
(\Theta + \delta\mathcal{B})|_{\partial\mathcal{M}} =& -\frac{\epsilon_{\partial\mathcal{M}}}{2\kappa_N^2}\left[\left(K^{MN} - \left(K - \frac{1}{\ell}\right)\gamma^{MN}\right)\delta G_{MN} + 8n^R\partial_R\Phi\,\delta\Phi\right] \\
&+ \frac{1}{2\kappa_N^2}\delta\left(e^{-8\Phi} \star H\right)B|_{\partial\mathcal{M}} + \mathrm{d}C\,,
\end{aligned}
\tag{9}
$$

where

$$C = c \cdot \epsilon_{\partial\mathcal{M}}\,, \qquad c^M = -\frac{1}{2\kappa_N^2}\gamma^{MN}n^R\delta G_{NR}\,. \tag{10}$$

In these expressions, extrinsic curvatures are computed with respect to the outward pointing unit normal $n = (\ell/r)\mathrm{d}r$, and $\gamma_{MN}$ refers to the metric induced at the boundary,

$$\gamma_{MN}\mathrm{d}x^M\mathrm{d}x^N = r^2\left(\eta_{ab} + \frac{\ell^2}{r^2}Y_{ab} + \dots\right)\mathrm{d}x^a\,\mathrm{d}x^b\,. \tag{11}$$

With the boundary conditions above, the term (9) behaves as

$$(\Theta + \delta\mathcal{B})|_{\partial\mathcal{M}} = \frac{1}{4\ell\kappa_N^2}\left(\ell^2\eta^{ab}\delta Y_{ab} + 16\delta\tilde{Y} + \dots\right)\mathrm{d}x^+ \wedge \mathrm{d}x^- + \mathrm{d}C\,, \tag{12}$$

where the total derivative on the boundary $\mathrm{d}C$ plays no role since its integral over $\partial\mathcal{M}$ vanishes. We thus define a well-posed variational problem if we demand

$$\tilde{Y} = -\frac{\ell^2}{16}\eta^{ab}Y_{ab} \equiv -\frac{\ell^2}{16}Y\,. \tag{13}$$

---

[2]One could think of generalizing the boundary conditions, allowing $C_0$ (and thus $\ell$) to vary in the spirit of [43]. While such an extension may be interesting, it is not needed for our purposes.

[3]The asymptotic value of the dilaton is also fixed in terms of $C_0$ as written in the boundary conditions in order to have non-trivial solutions. One would typically set it to 0 ($C_0 = 2$) by conformally rescaling the metric by a constant, but we will keep it general here.

Even though relating subleadings can in general be a dangerous restriction on a given space of solutions (potentially eliminating many or all of them), here we are justified in doing it because this is the condition required by conservation of the Kalb-Ramond charge,

$$\partial_M \left( e^{-8\Phi} \star H \right) = 0 \,, \tag{14}$$

and this is one of our equations of motion. Note that the last term in the boundary piece (7) converts to a fixed charge ensemble, allowing a well posed variational problem by fixing just $C_0$, with $b(x^a)$ free (without that term, we would need to demand $\delta b = 0$). The requirement to work at fixed charge is motivated by the fact that the leading term in the $B$-field in (5) is the one giving the charge, and not the pure gauge piece at the boundary, $b(x^a)$.[4] This is similar to the situation with standard gauge fields in low dimensions (e.g., AdS$_2$), where the non-normalizable (leading) mode specifies the charge instead of the chemical potential.[5]

We can now analyze the asymptotic symmetry transformations which preserve the previous boundary conditions. A general transformation is composed of a diffeomorphism and a gauge transformation of the $B$-field,

$$\delta_{\xi,\Lambda} G_{MN} = \mathcal{L}_\xi G_{MN} \,, \qquad \delta_{\xi,\Lambda} B_{MN} = \mathcal{L}_\xi B_{MN} + 2\partial_{[M}\Lambda_{N]} \,, \qquad \delta_{\xi,\Lambda}\Phi = \mathcal{L}_\xi \Phi \,. \tag{15}$$

Respecting the metric boundary conditions lands us in the Brown-Henneaux diffeomorphisms,

$$\xi[T^a] = -\frac{r}{2}\partial_a T^a \,\partial_r + \left( T^a + \frac{\ell^2}{2}\int^r \frac{\mathrm{d}r'}{(r')^3} h^{ab}\partial_b\partial_c T^c \right)\partial_a \,, \tag{16}$$

characterized by two chiral functions, $T^\pm(x^\pm)$. The subleadings in $\partial_a$ just ensure we stay in Fefferman-Graham gauge, and $h^{ab}$ denotes there the inverse of the (renormalized) metric induced at fixed $r$, $h_{ab} = \eta_{ab} + (\ell^2/r^2)Y_{ab} + \dots$. These diffeomorphisms also preserve the boundary conditions in the other fields,[6] thus they constitute asymptotic Killing vectors of our theory. Regarding transformations of the $B$-field, our boundary conditions allow asymptotically

$$\Lambda = \lambda + \mathcal{O}(r^{-1}) \,, \tag{17}$$

with $\lambda$ a one-form on fixed-$r$ surfaces. Here $\lambda$ just acts as a boundary gauge transformation, $b \to b + \mathrm{d}\lambda$.

The equations of motion (3) can be solved perturbatively in the asymptotic expansion. The $B$-field equation is simply charge conservation, $e^{-8\Phi} \star H = -C_0/\ell$. The dilaton equation of motion demands

$$\eta^{ab} Y_{ab} = 0 \,, \tag{18}$$

and the metric one then sets $\partial_- Y_{++} = \partial_+ Y_{--} = 0$. We thus obtain the standard two chiral functions $Y_{++} = Y_{++}(x^+)$, $Y_{--} = Y_{--}(x^-)$ of the Brown-Henneaux phase space. They transform under the asymptotic diffeomorphisms as

$$\delta_\xi Y_{\pm\pm} = T^\pm \partial_\pm Y_{\pm\pm} + 2Y_{\pm\pm}\partial_\pm T^\pm - \frac{1}{2}\partial_\pm^3 T^\pm \,. \tag{19}$$

---

[4] Incidentally, adding the boundary piece to work at fixed charge also makes the action finite for *any* configuration satisfying the boundary conditions (5). This would be a desirable feature in a hypothetical definition of a quantum gravity path integral.

[5] See [44] for a nice recent discussion about ensemble choices and natural boundary conditions in gravitational theories.

[6] This is true up to the fact that the diffeomorphisms generate $B_{ra}$ terms in the $B$-field, therefore taking us out of our chosen gauge. We must thus supplement the diffeomorphism by a compensating gauge transformation with parameter $\Lambda_\xi$ adapted to cancel these terms. This can be done and the needed $\Lambda_\xi$ is $\mathcal{O}(r^{-1})$, thus we neglect this technicality as it plays no role in what follows.

Charges are then computed using the covariant phase space analysis of [33, 45]. The general codimension-2 form relating charges between phase space solutions is

$$k_{BB} = \frac{\epsilon_{\partial\Sigma}}{\kappa_N^2} \tau_{[M} n_{N]} \left[ 2\xi^M \nabla^{[R} \delta G_R^{N]} + \xi^R \nabla^N \delta G_R^M + \frac{1}{2} \delta G_R^R \nabla^N \xi^M - \delta G_R^N \nabla^{[R} \xi^{M]} \right. \tag{20}$$

$$+ 8\delta\Phi\xi^N\nabla^M\Phi + \frac{e^{-8\Phi}}{4}\delta G_S^S H^{MNR} B_{RL}\xi^L + \frac{1}{2}\delta\left(e^{-8\Phi}H\right)^{MNR} B_{RL}\xi^L$$

$$\left. + e^{-8\Phi}\left(H^{RM[N}\xi^{S]}\delta B_{RS} - \delta B^{MR}G^{NS}\mathcal{L}_\xi B_{SR}\right)\right] - \frac{1}{2\kappa_N^2}\delta\left(e^{-8\Phi}\star H\right)\Lambda.$$

Here, $n_N$ is the radial unit normal, $\tau_M$ the unit normal to a Cauchy slice (orthogonal to $n_N$), and $\epsilon_{\partial\Sigma}$ the volume form at the boundary of a Cauchy slice (these satisfy $\epsilon = \tau \wedge n \wedge \epsilon_{\partial\Sigma}$). As usual, $\delta$ denotes a variation between different solutions of the phase space. The term within square brackets corresponds to diffeomorphism charges, while the final piece is the gauge charge (clearly integrable). Evaluating in our phase space of solutions and for the asymptotic symmetry transformations, all charges turn out to be integrable and we get

$$L[T] = \frac{\ell}{\kappa_N^2} \int_0^{2\pi} d\phi \left( Y_{++}\left(x^+\right) T^+\left(x^+\right) + Y_{--}(x^-)T^-(x^-) \right), \tag{21}$$

for diffeomorphism charges and

$$N[\Lambda] = -\frac{C_0}{2\ell\kappa_N^2} \int_0^{2\phi} d\phi \, \lambda_\phi, \tag{22}$$

for gauge transformations. For the gauge transformations we only get the global Kalb-Ramond charge (appearing for $\lambda_\phi = 1$, so $\Lambda = d\phi + \dots$),[7] but we get two Virasoro towers from the diffeomorphisms, since the algebra computed via

$$\{L[T_1], L[T_2]\} = \delta_{T_2}(L[T_1]), \tag{23}$$

is

$$i\left\{L_m^{(\pm)}, L_n^{(\pm)}\right\} = (m-n)L_{m+n}^{(\pm)} + \frac{\ell\pi}{\kappa_N^2} m^3 \delta_{m+n,0}, \tag{24}$$

where $L_m^{(\pm)}$ is the charge associated to the mode $T^\pm(x^\pm) = e^{imx^\pm}$. As expected, we have obtained the well-known Brown-Henneaux result (with the same central charge).

## 3 Chiral Brown-Henneaux and CSS as T-dual boundary conditions

In section 4, we will discuss how we can apply some notion of T-duality asymptotically to generate new boundary conditions from the ones of the previous section. Before doing so, however, we want to explore in this section a toy model of what T-duality can do to a certain set of boundary conditions and its associated solution space. In order to be as explicit as possible, we will work with a phase space which can be written in closed form, not only in an asymptotic expansion; and we will take it to have an exact Killing vector for all its field

---

[7]Clearly only the $\phi$-independent part of $\lambda_\phi$ gives charges, but one may ask why we do not allow $\lambda_\phi = \lambda_\phi(t)$. The reason is that charges are not conserved in that case, as it is easy to check. This can be understood as follows. Conservation requires (off-shell) invariance of the action, which due to the form of $\Theta$ demands $d\Lambda = 0$ at the boundary, so $\partial_\phi \lambda_t = \partial_t \lambda_\phi$. Expanding in modes, this equation forces the $\phi$-independent part of $\lambda_\phi$ to be constant.

configurations. These conditions can be met starting from the well-known Bañados phase space for AdS$_3$ gravity [41, 42],

$$ds^2 = \frac{\ell^2}{r^2}dr^2 - r^2\left(dx^+ - \frac{\ell^2}{r^2}Y_{--}(x^-)\,dx^-\right)\left(dx^- - \frac{\ell^2}{r^2}Y_{++}(x^+)\,dx^+\right). \tag{25}$$

In order to extend these metrics to solutions of our theory (1), we must supplement them with an appropriate dilaton and Kalb-Ramond two-form [7]. To make contact with the previous section, we choose

$$B = \left(\frac{r^2}{C_0} + b(x^a) + \frac{\ell^4}{C_0 r^2}Y_{++}\left(x^+\right)Y_{--}(x^-)\right)dx^+ \wedge dx^-, \qquad \Phi = \frac{1}{4}\log\left(\frac{2}{C_0}\right). \tag{26}$$

These are all solutions of our theory and they satisfy the boundary conditions (5), but notice that we are not claiming they are *all* the possible solutions satisfying such boundary conditions. In particular, matter fields being present, in the phase space of the previous section there will be backreacted solutions where the space is not everywhere locally AdS$_3$. These are not being considered now, so we are looking at a subset of the phase space in the previous section. As anticipated above, in order to apply T-duality we want to further restrict the field configurations so that we have an exact Killing vector throughout all of them, so we introduce a chiral version of the Bañados phase space in which we only keep left-moving excitations,

$$ds^2 = \frac{\ell^2}{r^2}dr^2 - r^2\left(dx^+ - \frac{\ell^2}{r^2}y_{--}\,dx^-\right)\left(dx^- - \frac{\ell^2}{r^2}Y_{++}(x^+)\,dx^+\right), \tag{27a}$$

$$B = \left(\frac{r^2}{C_0} + b\left(x^+\right) + \frac{\ell^4}{C_0 r^2}Y_{++}\left(x^+\right)y_{--}\right)dx^+ \wedge dx^-, \tag{27b}$$

$$\Phi = \frac{1}{4}\log\left(\frac{2}{C_0}\right), \tag{27c}$$

where lowercase $y_{--}$ emphasizes this is a constant now.

Since this is a subset of solutions within the phase space of the previous section, we can reuse most of the analysis done there to obtain the charges. Within the restricted phase space considered now, however, we will just get a single Virasoro tower, because we have frozen one of the chiral components. Let us be a bit more explicit. The asymptotic symmetry transformations are Brown-Henneaux diffeomorphisms (16) with $T^- = t^-$ a constant, as well as gauge transformations with asymptotic form $\lambda = \alpha\left(x^+\right)dx^- + \tilde{\lambda}$, with $\tilde{\lambda}$ a closed one-form on the boundary. Notice that the algebra of these transformations mixes non-trivially. Indeed, expanding in modes using $x^+ \sim x^+ + 2\pi$,

$$T_n = e^{inx^+}\partial_+ + \dots, \qquad \alpha_m = e^{imx^+}dx^- + \dots, \tag{28}$$

symmetry transformations $\delta_{(\xi,\Lambda)}$ form the algebra

$$i\left[\delta_{(T_m,0)}, \delta_{(T_n,0)}\right] = (m-n)\delta_{(T_{m+n},0)}, \tag{29a}$$

$$i\left[\delta_{(0,\alpha_m)}, \delta_{(0,\alpha_n)}\right] = 0, \tag{29b}$$

$$i\left[\delta_{(T_m,0)}, \delta_{(0,\alpha_n)}\right] = -n\delta_{(0,\alpha_{m+n})}. \tag{29c}$$

At the level of the symmetry transformations, we have just found a Witt tower semi-directly with a $\mathfrak{u}(1)$ loop algebra (plus the zero mode $\partial_-$). It is interesting to observe though that not all of these are real symmetries of our theory, since the charges associated with gauge transformations vanish (other than the global charge). The argument is the same presented

in the previous section: Except for the zero mode, charges of the form (22) integrate to zero. We are thus left with a single Witt algebra associated to the left-moving sector.

The phase space has now an exact Killing vector, $\partial_-$, along which we can perform a T-duality transformation.[8] The simplest way to do this is by first going to string frame via $\tilde{G}_{MN} = e^{4\Phi}G_{MN}$, and then applying the standard Buscher rules. To do so, consider the Killing vector normalized to get $R_\eta^2 = \tilde{G}_{MN}\eta^M\eta^N$ adimensional (pick $\eta = \ell^{-1}\partial_-$) and complete $\eta$ to a basis of the tangent space, $\{\eta, u_i\}$ with $i = 1, 2$. Defining the auxiliary field $M_{MN} = \tilde{G}_{MN} - B_{MN}$, Buscher rules give the dual solution in string frame as

$$\widehat{M}_{MN}u_i{}^M u_j{}^N = M_{MN}u_i{}^M u_j{}^N - \frac{1}{R_\eta^2}M_{MR}M_{SN}\eta^R\eta^S u_i{}^M u_j{}^N \,, \tag{30a}$$

$$\widehat{M}_{MN}\eta^M u_i{}^N = -\frac{1}{R_\eta^2}M_{MN}\eta^M u_i{}^N \,, \qquad \widehat{M}_{MN}u_i{}^M\eta^N = \frac{1}{R_\eta^2}M_{MN}u_i{}^M\eta^N \,, \tag{30b}$$

$$\widehat{M}_{MN}\eta^M\eta^N = \frac{1}{R_\eta^4} \,, \qquad\qquad\qquad e^{4\widehat{\Phi}} = \frac{1}{R_\eta^4}e^{4\Phi} \,, \tag{30c}$$

where the dual metric and $B$-field are read from the symmetric and antisymmetric parts of $\widehat{M}_{MN}$. Finally, we go back to Einstein frame with the transformation $\widehat{G}_{MN} = e^{-4\widehat{\Phi}}\widehat{\tilde{G}}$.

Dropping tildes from now on, the result of this procedure in the chiral restriction of the Bañados phase space produces as dual solution space

$$ds^2 = \frac{\hat{\ell}^2}{\rho^2}d\rho^2 - \rho^2 dx^+\left(dx^- - P\left(x^+\right)dx^+\right) + \hat{\ell}^2\left(L(x^+)(dx^+)^2 + \Delta\left(dx^- - P\left(x^+\right)dx^+\right)^2\right)$$

$$- \frac{\Delta\hat{\ell}^4}{\rho^2}L\left(x^+\right)dx^+\left(dx^- - P\left(x^+\right)dx^+\right) \,, \tag{31a}$$

$$B = \left(\frac{\rho^2}{\widehat{C}_0} + \frac{\Delta\hat{\ell}^4}{\widehat{C}_0}\frac{1}{\rho^2}L\left(x^+\right)\right)dx^+ \wedge dx^- \,, \tag{31b}$$

$$\Phi = \frac{1}{4}\log\left(\frac{2}{\widehat{C}_0}\right) \,, \tag{31c}$$

where we have rescaled the radial coordinate and defined some new constants and functions to ease the notation:

$$\rho = \sqrt{\frac{2y_{--}}{C_0}}r \,, \qquad \hat{\ell}^2 = \frac{4\ell^2 y_{--}^2}{C_0^2} \,, \qquad \widehat{C}_0 = \frac{4y_{--}^2}{C_0} \,,$$

$$\Delta = \frac{C_0^2}{4y_{--}} \,, \qquad P(x^+) = \frac{b\left(x^+\right)}{\ell^2} \,, \qquad L\left(x^+\right) = Y_{++}\left(x^+\right) \,. \tag{32}$$

In this form, the above metrics can be easily recognized as those forming the CSS phase space [38] (supplemented with the appropriate $B$-field and dilaton to turn them into solutions of the theory (1)), so we have just shown that the chiral Bañados and CSS phase spaces are T-dual. Notice that $\Lambda = -C_0/\ell^2 = -\widehat{C}_0/\hat{\ell}^2$ and, to make contact with the standard CSS analysis, we are treating $y_{--}$ (correspondingly, $\Delta$) as a *fixed* constant. So, for each chiral Bañados phase

---

[8]Technically, $\partial_-$ may not be a spacelike direction throughout the whole spacetime, and it is certainly null asymptotically. It is then likely that a proper worldsheet definition of T-duality along this direction is not available. However, in this work we regard T-duality as a solution-generating technique in (super)gravity, so we will just apply Buscher rules, taking advantage of the fact that they relate solutions with an isometry to new solutions, irrespective of the string theory definition of the transformation.

space with a certain value of $y_{--}$, we get a corresponding CSS phase space with fixed $\Delta$.[9]

Asymptotic symmetry transformations are diffeomorphisms generated by vector fields of the form

$$\xi[\epsilon,\sigma] = -\frac{\rho}{2}\epsilon'(x^+)\partial_\rho + (\epsilon(x^+) + \mathcal{O}(\rho^{-2}))\partial_+ + (\sigma(x^+) + \mathcal{O}(\rho^{-2}))\partial_-, \quad (33)$$

and we do not have asymptotic gauge transformations of the $B$-field non-vanishing at the boundary because the duality has not produced a fluctuating $\mathcal{O}(\rho^0)$ term. The vector fields form a Witt algebra together with an Abelian loop algebra. In fact, the algebra is exactly (29), where $T_m$ are modes of $\epsilon(x^+)$ and $\alpha_m$ are now modes of $\sigma(x^+)$, thus coming from diffeomorphisms instead of gauge transformations.[10] This is a manifestation of the fact that gauge transformations of the $B$-field become diffeomorphisms after T-duality, and viceversa (as can be directly checked from Buscher rules (30)). It is intriguing however to note that now all transformations are real symmetries, since the charges computed using the covariant phase space method (20) become

$$\mathcal{L}[\epsilon] = \frac{\hat{\ell}}{\kappa_N^2} \int_0^{2\pi} d\phi\, \epsilon(x^+)(L(x^+) - \Delta P^2(x^+)), \quad (34a)$$

$$\mathcal{N}[\sigma] = 2\Delta \frac{\hat{\ell}}{\kappa_N^2} \int_0^{2\pi} d\phi\, \sigma(x^+) P(x^+), \quad (34b)$$

where, contrary to [38], we have not shifted the zero mode charge of $\sigma$. Using appropriate versions of (23), the algebra of charges becomes

$$i\{\mathcal{L}_m, \mathcal{L}_n\} = (m-n)\mathcal{L}_{m+n} + \frac{\hat{\ell}\pi}{\kappa_N^2} m^3 \delta_{m+n,0}, \quad (35a)$$

$$i\{\mathcal{N}_m, \mathcal{N}_n\} = -4\Delta \frac{\hat{\ell}\pi}{\kappa_N^2} m\, \delta_{m+n,0}, \quad (35b)$$

$$i\{\mathcal{L}_m, \mathcal{N}_n\} = -n\mathcal{N}_{n+m}, \quad (35c)$$

where we have again expanded in modes $\epsilon_m(x^+) = e^{imx^+}$, $\sigma_m(x^+) = e^{imx^+}$.

Let us end this section with a brief summary and some comments about the result obtained. By restricting the phase space of solutions to a chiral subset possessing an exact Killing vector, (27), we can apply a T-duality transformation to all such solutions, obtaining an exact notion of T-dual phase space. This procedure has uncovered the surprising result that one (chiral) half of the Brown-Henneaux phase space is T-dual to the CSS phase space. Perhaps even more surprisingly, the algebra of symmetry transformations is the same in both cases, but this is not the case for the actual algebra of non-trivial charges. This can be traced back to the exchange between $B$-field gauge transformations and diffeomorphisms that T-duality produces. The Abelian tower comes from gauge transformations in the chiral Brown-Henneaux case, and most of these are trivial (i.e., they have vanishing charge). That same tower arises from diffeomorphisms in CSS which have non-trivial charges. This difference is particularly puzzling if we think of T-duality not as a solution generating technique in (super)gravity, but as an

---

[9]One could try to allow $\Delta$ to vary following the argument presented in the appendix of [38]. However, defining a phase space for all values of $y_{--}$ and dealing with the whole set of duals still presents problems, because we would be varying also the $B$-field charge $\hat{C}_0$ and AdS radius $\hat{\ell}$. We thus keep the example simple and discuss the duality at fixed $y_{--}$ and $\Delta$.

[10]The invariance under T-duality of the asymptotic symmetry transformations we are observing here can be proven in full generality (without reference to any specific background, just assuming there is a $U(1)$ symmetry to dualize) by using the form of Buscher rules, as shown in appendix A.

actual equivalence of stringy origin. In that philosophy, we would expect that the chiral Brown-Henneaux and CSS backgrounds define equivalent theories, and the symmetry algebras should then match. Of course, in a stringy setup one could question the consistency of unnaturally chopping off the allowed set of solutions to just a chiral half of the set of all asymptotically AdS$_3$ metrics (after all, the excluded backgrounds should also be valid stringy excitations), so we refrain from giving too much relevance to the mismatch in the toy model we just presented.

# 4 Dual boundary conditions: A phase space for the black string

After having discussed how T-duality may affect the asymptotic structure of a theory in a controlled and simple setup, we aim now to use the well-understood Brown-Henneaux boundary conditions (5), together with some asymptotic notion of T-duality, to generate a new set of boundary conditions. At the very least, this will allow us to obtain boundary conditions defining a phase space which contains the three-dimensional black strings. Indeed, black strings are T-dual to BTZ black holes [7]. Since BTZ black holes are included within the Brown-Henneaux phase space, the dual black strings will be included in any notion of T-dual phase space we are able to define. More broadly, we believe the construction can serve as a blueprint for how to generate new boundary conditions using T-duality. This is interesting in its own right, since in many cases T-duality heavily affects the asymptotic structure, and thus we can use well understood boundary conditions (here, Brown-Henneaux AdS$_3$ asymptotics) to generate novel ones (here, the ones containing black strings, which are asymptotically flat in some sense to be made precise below).

## 4.1 T-dual boundary conditions

We start our journey from (5). These boundary conditions possess

$$\eta = \frac{\partial_+ - \partial_-}{\ell} = \frac{\partial_\phi}{\ell} \, , \tag{36}$$

as an exact Killing vector of the leading components. This is the condition we demand to apply our asymptotic notion of T-duality, expecting this will produce dual fields whose leading components also provide a solution (the situation is actually slightly subtler, as we will shortly discuss). A direct application of Buscher rules (30) gives us the following dual boundary conditions (in Einstein frame):

$$ds^2 = \left( 1 + \frac{F(x^a)}{\hat{r}} + \dots \right) d\hat{r}^2 + \hat{r}^2 \left( M_{ab} + \frac{1}{\hat{r}} Z_{ab} + \dots \right) dx^a \, dx^b \, , \tag{37a}$$

$$B = \frac{1}{2} \left( \frac{\tilde{\beta}(x^a)}{\hat{r}} + \dots \right) dz \wedge dw \, , \tag{37b}$$

$$e^{4\Phi} = \frac{\hat{r}_0^2}{\hat{r}^2} \left( 1 + \frac{\psi(x^a)}{\hat{r}} + \dots \right) , \tag{37c}$$

where, to simplify the notation, we have redefined our coordinates as

$$\hat{r} = \frac{r^2}{C_0 \ell} \, , \qquad \frac{z}{\sqrt{C_0}} = x^+ + x^- = 2\frac{t}{\ell} \, , \qquad \frac{w}{\sqrt{C_0}} = -x^+ + x^- = -2\phi \, , \tag{38}$$

and $\hat{r}_0 = \ell/\sqrt{2C_0}$. New fluctuations are related to the old ones in a way which will not be very relevant for us, e.g.,

$$F = \frac{2\ell}{C_0} \left( Y_{++} + Y_{--} - Y_{+-} \right) , \qquad \psi = -\frac{2\ell}{C_0} \left( Y_{++} + Y_{--} - \frac{3}{2} Y_{+-} \right) , \qquad \tilde{\beta} = \frac{\ell^3}{2C_0^2} \left( Y_{++} - Y_{--} \right) , \tag{39}$$

and these are all functions of $(x^a) = (z, w)$. However, two facts that naturally come out of the duality are crucial. The first one is the form of the fluctuating leading metric $M_{ab}$,

$$(M_{ab}) = \begin{pmatrix} A(x^a) & -1/2 \\ -1/2 & 0 \end{pmatrix}, \qquad A = \frac{2b}{\ell^2} + \frac{2Y_{+-}}{C_0}. \tag{40}$$

Notice that subleading terms in the original boundary conditions became part of the leading pieces after dualizing, in particular through $A(z, w)$. Thus, it is not true that the leading pieces of (37) provide a valid solution, and going on-shell will later impose non-trivial restrictions on $A(z, w)$ – see (58). The other important fact is that $Z_{ww}$, the leading piece in $dw^2$ (since $M_{ww} = 0$), is fixed to be a constant

$$(Z_{ab}) = \begin{pmatrix} Z_{zz}(x^a) & Z_{zw}(x^a) \\ Z_{zw}(x^a) & \ell/4 \end{pmatrix}. \tag{41}$$

$Z_{zz}$ and $Z_{zw}$ can be obtained as combinations of the old fluctuations from the Buscher transformations, although it will not be needed in the following discussion. $Z_{zw}$ is found to be linear in the fluctuations $Y_{ab}$, while $Z_{zz}$ is quadratic in $Y_{ab}$ and includes also further subleading pieces from the Brown-Henneaux boundary conditions, (5). This different behavior can be traced back to the manifestly different treatment Buscher rules do of the dualizing coordinate ($w$ or $\phi$) versus the remaining directions.

As promised, the three-dimensional black string solutions introduced in [18] can be shown to satisfy these boundary conditions. However, due to the nature of our construction, they do so in their form obtained by T-dualizing the BTZ black holes [7]. We refer the reader to appendix B for details and we just quote here the main results. BTZ black holes correspond to solutions with constant $Y_{++} = L_+$ and $Y_{--} = L_-$ in the boundary conditions (5). Buscher rules transform them to

$$\begin{aligned} ds^2 = g^4(\hat{r}) d\hat{r}^2 + \hat{r}^2 g^2(\hat{r}) \Bigg[ \left( \frac{2b}{\ell^2} + \frac{4\ell L_+ L_-}{C_0^2 \hat{r}} \right) \left( 1 + \frac{b}{2\ell\hat{r}} \right) dz^2 + \frac{\ell}{4\hat{r}} dw^2 \\ - \left( 1 + \frac{b}{\ell\hat{r}} + \frac{\ell^2 L_+ L_-}{C_0^2 \hat{r}^2} \right) dz dw \Bigg], \end{aligned} \tag{42a}$$

$$B = \frac{\ell^3 (L_+ - L_-)}{4 C_0^2 \hat{r} g^2(\hat{r})} dz \wedge dw, \tag{42b}$$

$$e^{4\Phi} = \frac{\hat{r}_0^2}{\hat{r}^2 g^4(\hat{r})}, \tag{42c}$$

where we are using coordinates (38), and $g^2(\hat{r})$ is

$$g^2(\hat{r}) = 1 + \frac{\ell}{C_0 \hat{r}} (L_+ + L_-) + \frac{\ell^2}{C_0^2 \hat{r}^2} L_+ L_-. \tag{43}$$

Note also that $b = b(t) = b(z)$, since any dependence on $\phi$ is forbidden by the requirement to have an exact angular Killing vector in the BTZ background, needed to apply T-duality. All these solutions satisfy the dual boundary conditions (37), and they do so for constant subleading pieces satisfying $F = -\psi$. One can take these backgrounds to the more familiar black string form via the coordinate changes

$$dw = dW + \left( \frac{2b(z)}{\ell^2} - \frac{2(L_+ + L_-)}{C_0} \right) dz, \tag{44}$$

and

$$R = \hat{r} g^2(\hat{r}), \qquad z = \frac{\sqrt{C_0}(T+X)}{2(L_+ L_-)^{1/4}}, \qquad W = \frac{(\sqrt{L_+}+\sqrt{L_-})^2 T + (\sqrt{L_+}-\sqrt{L_-})^2 X}{\sqrt{C_0}(L_+ L_-)^{1/4}}, \quad (45)$$

after which the solution becomes

$$ds^2 = \frac{dR^2}{\left(1-\frac{\mathcal{M}}{R}\right)\left(1-\frac{\mathcal{Q}^2}{\mathcal{M}R}\right)} + R^2\left[-\left(1-\frac{\mathcal{M}}{R}\right)dT^2 + \left(1-\frac{\mathcal{Q}^2}{\mathcal{M}R}\right)dX^2\right], \qquad (46a)$$

$$B = -\frac{\hat{r}_0^2 \mathcal{Q}}{R} dT \wedge dX, \qquad (46b)$$

$$e^{4\Phi} = \frac{\hat{r}_0^2}{R^2}, \qquad (46c)$$

with

$$\mathcal{M} = \frac{\ell}{C_0}\left(\sqrt{L_+}+\sqrt{L_-}\right)^2, \qquad \mathcal{Q}^2 = \frac{\ell^2}{C_0^2}(L_+ - L_-)^2. \qquad (47)$$

Note also that the solution with all fluctuations turned off in the boundary conditions (37) (i.e., the dual of a massless BTZ black-hole) is

$$ds^2 = d\hat{r}^2 + \hat{r}^2\left(-dz\,dw + \frac{\ell}{4\hat{r}}dw^2\right), \qquad (48)$$

which is a plane wave in the presence of a dilaton $e^{4\Phi} = \hat{r}_0^2/\hat{r}^2$. The fact that this cannot be mapped to the $\mathcal{M} = \mathcal{Q} = 0$ black string was already noticed in [7], and the reason can be traced back to the transformation (45) being ill-defined for extremal solutions with either $L_+$ or $L_-$ vanishing.[11] Therefore, the precise statement is that our boundary conditions include all solutions of the form (42), many of which can be mapped to non-extremal black strings in the standard form (46). Extremal black strings, defined by setting $\mathcal{M} = |\mathcal{Q}|$ in (46), are not part of our configuration space.

It is interesting to note that our boundary conditions (37) share some similarities with the CSS metrics discussed in the previous section, (31). In particular, the coordinate in which we dualize ($x^-$ there, $w$ here) has a vanishing leading piece in the metric, and the first subleading is fixed ($\hat{\ell}^2\Delta$ there, $\ell/4$ here). This is the first of many similarities with CSS we will encounter, but note that the different asymptotic behaviour of the Killing vector used to dualize ($R_\eta^2 \sim r^2$ now) has completely changed the asymptotic curvature. In particular, we are no longer dealing with locally AdS$_3$ metrics, as shown by the large-$\hat{r}$ expansion of the Ricci scalar

$$R = \frac{\mathcal{R}-2}{\hat{r}^2} + \mathcal{O}(\hat{r}^{-3}), \qquad (49)$$

where $\mathcal{R} = 4\partial_w^2 A$ is the Ricci scalar of the 2d metric $M_{ab}$. The boundary conditions are asymptotically flat in the sense that this curvature scalar decays as $\hat{r}^{-2}$, but (37) are not standard asymptotically flat boundary conditions (in particular, for $A = 0$, the Killing vectors $\partial_z \pm \partial_w$ have diverging norm at large $\hat{r}$). It is also important to keep in mind for the forthcoming analysis that the fluctuating $A(x^a)$ forbids a clear identification of timelike and spacelike directions in the asymptotic region. Even though $z$ comes from the time $t$, we have $\partial_z^2 \sim \hat{r}^2 A$, so $z$ is actually a spacelike direction if $A > 0$. We will come back to this point later when integrating charges, since it will be relevant to pick our Cauchy surface of integration.

---

[11]Another notable exceptional case is the solution obtained by dualizing empty AdS$_3$, for which the transformation (45) also does not make sense. Such dual, included in the boundary conditions (37), is a product of time and the two-dimensional Euclidean black hole [7, 46].

Let us briefly discuss the variational problem with the boundary conditions (37). The boundary term (7) must be slightly modified to

$$\hat{\mathcal{B}} = \frac{1}{\kappa_N^2} \left( K - \frac{e^{2\Phi}}{\hat{r}_0} \right) \epsilon_{\partial\mathcal{M}}, \tag{50}$$

so that varying the action with this modified boundary term produces

$$\left( \Theta + \delta\hat{\mathcal{B}} \right)\big|_{\partial\mathcal{M}} = -\frac{\epsilon_{\partial\mathcal{M}}}{2\kappa_N^2} \left[ \left( K^{MN} - \left( K - \frac{e^{2\Phi}}{\ell} \right) \gamma^{MN} \right) \delta G_{MN} + 8 \left( n^R \partial_R \Phi + \frac{e^{2\Phi}}{2\hat{r}_0} \right) \delta\Phi \right] + \mathrm{d}C, \tag{51}$$

with $\mathrm{d}C$ as in (9) an irrelevant total derivative.[12] Using the boundary conditions gives

$$\left( \Theta + \delta\hat{\mathcal{B}} \right)\big|_{\partial\mathcal{M}} = \frac{\ell}{8\kappa_N^2} \delta A \, \mathrm{d}z \wedge \mathrm{d}w + \mathrm{d}C. \tag{52}$$

This structure is exactly the same appearing in the CSS construction [38]: The non-vanishing contribution can be written as $Z_{ab}\delta M^{ab}$, with $M^{ab}$ the inverse of $M_{ab}$. In order to have a well-defined variational problem, we thus apply the same trick presented in [38] and include an extra, non-covariant piece in the action

$$S \longrightarrow S - \frac{1}{\kappa_N^2} \int_{\partial\mathcal{M}} \mathrm{d}^2x \, \sqrt{-\det M_{ab}} \, \frac{\ell}{4} A(z), \tag{53}$$

such that now $\delta S = 0$ with the boundary conditions (37).

## 4.2  Asymptotic symmetry transformations and charges

The asymptotic symmetry transformations preserving the boundary conditions (37) are diffeomorphisms generated by

$$\begin{aligned}
\xi[R,Q,T,S] = {} & \ell \left( f(z,w) + \dots \right) \partial_{\hat{r}} + \left( T(z) - \frac{2\ell}{\hat{r}} \partial_w f + \dots \right) \partial_z \\
& + \left( S(z) - wT'(z) - \frac{2\ell}{\hat{r}} \left( \partial_z f + 2A\partial_w f \right) + \dots \right) \partial_w,
\end{aligned} \tag{54}$$

where we have written the leading $\hat{r}$ component as

$$f(z,w) = R(z) + wQ(z) + \frac{w^2}{8} T'(z), \tag{55}$$

and the form of $\xi^{\hat{r}}$ comes from imposing $\delta Z_{ww} = 0$. Since we are now fixing the boundary value of the $B$-field, the allowed gauge transformations have $\Lambda = \tilde{\lambda} + \mathcal{O}(\hat{r}^{-1})$ with $\tilde{\lambda}$ a closed one-form on the boundary. Much like with Brown-Henneaux boundary conditions, this will only produce the global charge, so we will not discuss it further.

Before computing the charges associated to these transformations and the corresponding symmetry algebra, we need to go on-shell and impose the conditions derived from the equations of motion, (3). From the dilaton equation of motion (note that the cosmological constant is $\Lambda_0 = -C_0/\ell^2 = -1/(2\hat{r}_0^2)$)

$$F + 2\psi + Z + M^{ab} D_a D_b \psi = 0, \tag{56}$$

---

[12]It would be interesting to find a T-duality invariant boundary term able to produce a well-defined variational problem with both boundary conditions, (5) and (37). Also, we have dropped the piece in $\mathcal{B}$ fixing the charge, $(e^{-8\Phi} \star H)B$. While it can be included, we believe it is more natural to think of the boundary conditions (37) in a "grand canonical" ensemble, with $B$ fixed to zero at the boundary. One could actually add to (37) a piece $B_{zw} = (\tilde{b} + \tilde{\beta}/\hat{r} + \dots)/2$, such that $B$ goes to a fixed $\tilde{b}$ at the boundary. The following analysis still holds, so we prefer to leave $\tilde{b} = 0$ as this value naturally comes out of the duality.

where $D_a$ is the covariant derivative associated to the 2d metric $M_{ab}$, and $Z = M^{ab}Z_{ab}$. The $B$-field equation simply requires to have constant charge, so $\partial_M \tilde{\beta} = 0$. Finally, the metric equation of motion gives at leading order in the radial part the requirement that $M_{ab}$ must be locally flat

$$\mathcal{R} = 4\partial_w^2 A = 0 \,, \tag{57}$$

so we get

$$A(z,w) = A_0(z) + wA_1(z) \,. \tag{58}$$

Subleading pieces give the following three conditions

$$D^2 Z - D^a D^b Z_{ab} = F + 2\psi + Z \,, \tag{59a}$$

$$M^{cd} D_c Z_{da} = \partial_a (F + \psi + Z) \,, \tag{59b}$$

$$D_a D_b F = (D^2 F) M_{ab} \,, \tag{59c}$$

where indices in derivatives $D_a$ were raised with $M^{ab}$. Solving all these equations in full generality is fortunately not needed for our purposes. We can just impose them as conditions when we are evaluating on-shell quantities, such as the charges we are about to compute. In particular, the explicit form of the second equation in (59) is going to be relevant. The $z$-component is

$$2\partial_w (Z_{zz} + 2AZ_{zw}) = -\partial_z \left( F + \psi - 2Z_{zw} - \frac{\ell}{2}A \right) \,, \tag{60}$$

while the $w$-component imposes

$$\partial_w \left( F + \psi - 2Z_{zw} + \frac{\ell}{2}A \right) = 0 \,. \tag{61}$$

There is a final point we need to address before obtaining the charges associated with the asymptotic symmetry transformations (54): We need to fix the Cauchy slice used to compute them (at least asymptotically close to the boundary, where charges will be evaluated). Naively, we could think that surfaces of constant $z$ are the natural choice, since $z$ is directly related to $t$ before the duality transformation. However, as indicated before, these surfaces are asymptotically null, so not really well-suited for the standard Hamiltonian analysis. Since we are using T-duality merely as a solution generating technique, we will not relate the choices after the transformation with those made before: We regard the theory with the boundary conditions (37) as a well defined entity of its own, not tied to the structure before the duality. The form of the boundary conditions suggests a better choice of Cauchy slices: Those with constant $w$, so that we will perform integrals over $z$. Compactifying this spatial coordinate to regulate IR issues, we will be able to expand functions such as $A_0(z)$ in Fourier modes, as it is conventional.

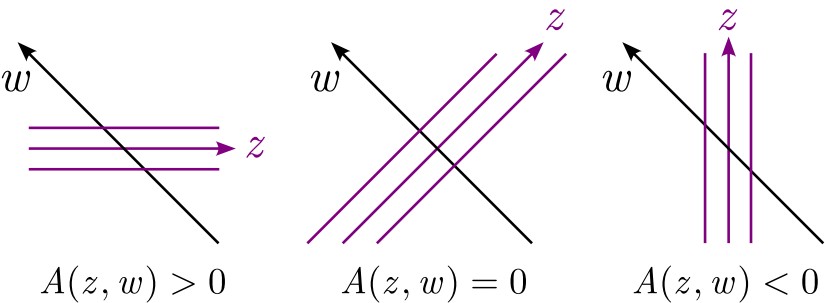

Figure 1: In purple, constant $w$ surfaces at the asymptotic boundary depending on the sign of $A(z,w)$. We take our bulk Cauchy slices to meet the boundary at these surfaces, so that charges are integrated over $z$.

Note that there are still potential issues when $A(z,w) < 0$, since in that case constant $w$ surfaces are asymptotically timelike (alternatively, $\partial_z^2 \sim \hat{r}^2 A(z,w)$ becomes negative, see figure 1). Nevertheless, a choice must be made, because having a fluctuating leading boundary metric given by $M_{ab}$ forbids the identification of spacelike surfaces consistently throughout the whole phase space, and we believe the results that will be presented below justify the choice of constant $w$ surfaces as surfaces of integration. Incidentally, it is worth mentionning that this is also very similar to the choice made in the CSS construction [38]. There, the equivalent to $(z,w)$ coordinates are light-cone coordinates on the boundary, and Cauchy slices are taken to have $z + w =$ constant. It is straightforward to generalize our discussion to this alternative choice of Cauchy slice, and our results still hold. But it must be noted that also in CSS the leading boundary metric has a fluctuating piece, $P(x^+)$ using our notation in (31), and the chosen $2t = x^+ + x^- =$ constant surfaces are spacelike if and only if $P(x^+) > -1$.

We can now compute the charges associated with the asymptotic symmetry transformations. Using the general form presented in (20), we obtain integrable diffeomorphism charges of the form:

$$\mathcal{T} = \frac{1}{2\kappa_N^2} \int \mathrm{d}z \left[ (Z_{zz} + 2A Z_{zw}) T - \frac{w}{2} \left( F + \psi - 2 Z_{zw} - \frac{\ell}{2} A_0 \right) T' \right], \tag{62a}$$

$$\mathcal{S} = \frac{1}{4\kappa_N^2} \int \mathrm{d}z \left( F + \psi - 2 Z_{zw} + \frac{\ell}{2} A \right) S, \tag{62b}$$

$$\mathcal{R} = -\frac{\ell}{\kappa_N^2} \int \mathrm{d}z A_1 R, \tag{62c}$$

$$\mathcal{Q} = \frac{\ell}{\kappa_N^2} \int \mathrm{d}z A_0 Q, \tag{62d}$$

where calligraphic letters denote the charges associated with $T(z)$, $S(z)$, $R(z)$, and $Q(z)$. Verifying that these charges are conserved (i.e., they do not depend on the fixed value of $w$ used for the integrals) is a very non-trivial consistency check of our result. This is obvious for $\mathcal{R}$ and $\mathcal{Q}$, since they do not depend on $w$ at all.[13] For $\mathcal{S}$, the implicit dependence through the phase space functions combines to give a $w$-independent quantity on-shell, as equation (61) shows. For $\mathcal{T}$, the implicit and explicit dependences can be shown to cancel by virtue of equations (60), (61), and the fact that total derivatives integrate to zero over the full boundary cycle.

The symmetry algebra follows from (23), properly adapted to the charges just obtained. Appendix C contains auxiliary results to evaluate the variation of the phase space functions under the action of the asymptotic Killing vectors. Using them and expanding in modes the functions appearing in the vectors,[14] we get

$$i\{\mathcal{T}_m, \mathcal{T}_n\} = (m-n)\mathcal{T}_{m+n}, \qquad i\{\mathcal{T}_m, \mathcal{S}_n\} = -(m+n)\mathcal{S}_{m+n} + \frac{1}{4} m \mathcal{Q}_{m+n}, \tag{63a}$$

$$i\{\mathcal{T}_m, \mathcal{Q}_n\} = (m-n)\mathcal{Q}_{m+n}, \qquad i\{\mathcal{T}_m, \mathcal{R}_n\} = -n\mathcal{R}_{m+n} - i\frac{2\pi\ell}{\kappa_N^2} m^2 \delta_{m+n,0}, \tag{63b}$$

$$i\{\mathcal{S}_m, \mathcal{S}_n\} = -\frac{\pi\ell}{2\kappa_N^2} m \delta_{m+n,0}, \qquad i\{\mathcal{S}_m, \mathcal{Q}_n\} = i\mathcal{R}_{m+n} - \frac{2\pi\ell}{\kappa_N^2} m \delta_{m+n,0}, \tag{63c}$$

---

[13]It has been recently proposed that asymptotic charges can be classified into dynamical or kinematical [47], depending on whether or not they obey flux-balance laws. Dynamical and kinematical charges can behave very differently under a change of gauge. The trivial conservation of $\mathcal{R}$ and $\mathcal{Q}$ may be a sign that they are, in this language, kinematical. It would be interesting to explore this further.

[14]Mode expansions are $S_m \sim e^{imz} \partial_w$ and similarly for the other asymptotic Killing vectors. Note that this sets the periodicity $z \sim z + 2\pi$. This is not a restriction since rescaling $z$ can be mapped to a rescaling of the parameters in our boundary conditions.

with the remaining brackets vanishing,

$$\{\mathcal{S}_m, \mathcal{R}_n\} = \{\mathcal{R}_m, \mathcal{R}_n\} = \{\mathcal{R}_m, \mathcal{Q}_n\} = \{\mathcal{Q}_m, \mathcal{Q}_n\} = 0 \,. \tag{64}$$

In order to interpret this algebra, note that the $\mathcal{T}_m$ generate a Witt tower (i.e., a Virasoro algebra with $c = 0$). The remaining charges can be associated with fields of definite weight if we replace $\mathcal{S}_m$ by

$$\bar{\mathcal{S}}_m = \mathcal{S}_m - \frac{1}{8}\mathcal{Q}_m \,, \tag{65}$$

in which case brackets with $\mathcal{T}_m$ become

$$i\{\mathcal{T}_m, \mathcal{Q}_n\} = (m - n)\mathcal{Q}_{m+n} \,, \tag{66a}$$

$$i\{\mathcal{T}_m, \mathcal{R}_n\} = -n\mathcal{R}_{m+n} - i\frac{2\pi\ell}{\kappa_N^2}m^2\delta_{m+n,0} \,, \tag{66b}$$

$$i\{\mathcal{T}_m, \bar{\mathcal{S}}_n\} = -(m + n)\bar{\mathcal{S}}_{m+n} \,. \tag{66c}$$

$\mathcal{Q}$, $\mathcal{R}$ and $\bar{\mathcal{S}}$ have thus weight 2, 1, and 0 respectively; and we get a central extension between $\mathcal{T}$ and $\mathcal{R}$. After the redefinition we have $i\{\bar{\mathcal{S}}_m, \bar{\mathcal{S}}_n\} = 0$, so the only remaining non-trivial bracket is

$$i\{\bar{\mathcal{S}}_m, \mathcal{Q}_n\} = i\mathcal{R}_{m+n} - \frac{2\pi\ell}{\kappa_N^2}m\delta_{m+n,0} \,. \tag{67}$$

The generators $\{\bar{\mathcal{S}}_m, \mathcal{Q}_m, \mathcal{R}_m\}$ themselves form a subalgebra which can be identified by looking at the zero modes, for which the only non-vanishing bracket is

$$i\{\bar{\mathcal{S}}_0, \mathcal{Q}_0\} = i\mathcal{R}_0 \,. \tag{68}$$

The Hermitian generators $\{\bar{\mathcal{S}}_0, \mathcal{Q}_0, \mathcal{R}_0\}$ thus satisfy the commutation relations of the three-dimensional Heisenberg algebra, with $\bar{\mathcal{S}}_0$ and $\mathcal{Q}_0$ acting as "position" and "momentum" operators, and $\mathcal{R}_0$ being the central element. It is possible to build a loop algebra on top of this, following the standard procedure [48]. The algebra of the $\bar{\mathcal{S}}$, $\mathcal{Q}$ and $\mathcal{R}$ towers is a central extension of the result of this construction, namely

$$i\{J_m^a, J_n^b\} = if^{ab}{}_c J_{m+n}^c + \frac{2\pi\ell}{\kappa_N^2}g^{ab}m\delta_{m+n,0} \,, \tag{69}$$

where $a, b, c \in \{\bar{\mathcal{S}}, \mathcal{Q}, \mathcal{R}\}$ label the different towers of generators, $f^{ab}{}_c$ are the structure constants of the Heisenberg algebra (68) ($f^{\bar{\mathcal{S}}\mathcal{Q}}{}_{\mathcal{R}} = -f^{\mathcal{Q}\bar{\mathcal{S}}}{}_{\mathcal{R}} = 1$), and $g^{\bar{\mathcal{S}}\mathcal{Q}} = g^{\mathcal{Q}\bar{\mathcal{S}}} = -1$ are the non-zero components of $g^{ab}$. Notice that $g^{ab}$ is *not* the Killing metric of the three-dimensional Heisenberg algebra (which would be trivial). This is a consequence of the algebra (68) not being semisimple: In those cases, the central extension is not necessarily proportional to the Killing metric (e.g., a $\mathfrak{u}(1)$ algebra has a trivial Killing metric, but the loop algebra built from it admits a non-trivial central extension).

To summarize, the asymptotic symmetry algebra derived from the boundary conditions (37) has the form of a Witt tower (without central extension) plus three towers of weights two, one and zero respectively, which together form a central extension of the loop algebra constructed from the Heisenberg algebra (69). This is a very large algebra which contains some well-known subalgebras within it. The $\{\mathcal{T}, \mathcal{Q}\}$ generators form a $\mathfrak{bms}_3$ subalgebra without any central extension [49]; similarly, the $\{\mathcal{T}, \bar{\mathcal{S}}\}$ give a non-centrally extended $\mathfrak{bms}_2$ [50, 51]. Finally, the $\{\mathcal{T}, \mathcal{R}\}$ assemble together into a centrally extended version of the warped Witt algebra [5], where the central extension is trivial in the $\{\mathcal{P}_m, \mathcal{P}_n\}$ but non-trivial in the mixed bracket, and is thus often referred to as twisted warped Witt algebra [50, 52].

# 5 Discussion

From a (super)gravity perspective, T-duality is a transformation that takes as input solutions to the field equations and spits out new solutions, possibly with wildly different asymptotic behavior. A paradigmatic example of this is the duality between asymptotically AdS$_3$ BTZ black holes and black strings, which have asymptotically vanishing curvature. Strictly speaking, the duality transformation can only be used whenever the backgrounds have an exact Killing vector. However, by means of the BTZ / black string example, our goal in this paper has been to show that whenever we have a well-understood phase space in which one set of solutions can be embedded (namely, BTZ black holes in the Brown-Henneaux phase space of section 2), T-duality transformations can inform the construction of a dual phase space that includes the dual solutions (black strings in the phase space of section 4). In this sense, we could call a construction along the lines of the one presented in this paper *asymptotic T-duality*. Notice that this procedure does not directly relate a phase space of solutions with its dual. As we exemplified in section 4, what T-duality can do is generate a set of boundary conditions from existing ones, but the construction of a consistent phase space has to be done in the usual way starting from those new boundary conditions.

The construction is intended to be meaningful only at the level of classical gravitational theories. Thus, we view it as a way to generate new boundary conditions from existing ones, in a process that can eventually lead to interesting and new asymptotic symmetry algebras.[15] The different algebras make manifest that no kind of equivalence is in general expected between the theory with the original and the dual boundary conditions. This contrasts with the situation in string theory whenever an exact isometry is present, since then backgrounds related by T-duality are known to define equivalent theories [29].

One of the main byproducts of our analysis has been the construction of a phase space containing the three-dimensional black strings of Horne and Horowitz [18], at least whenever they are away from the extremal limit. Previous works have also addressed the problem of constructing such a phase space [36, 37], but our proposal resulting from asymptotic T-duality is different and new. We obtain a much larger symmetry algebra (66)-(67), potentially allowing more control over the set of states in the phase space. As a downside, the present construction does not allow us to discuss black strings in the extremal limit, since these are not obtained by dualizing BTZ black holes.

It would be interesting to further explore the implications of the symmetry algebra (66)-(67) for the theory with boundary conditions (37). A quantum gravitational theory with those boundary conditions would provide a representation of the aforementioned symmetry algebra on its Hilbert space, thus the study of the representations of the algebra could inform us about the possible spectrum of such a theory. Another interesting avenue to extract consequences from the symmetry algebra would be to derive some Cardy-like formula [53] able to constrain the density of states at sufficiently high energies. If such a formula exists, it should reproduce the entropy of the black strings as derived in appendix B, thus giving a microscopic, symmetry-based argument for its origin. Since it is simple enough to be suggestive, let us reproduce here the result (B.24) giving the entropy of the black strings (42) in terms of their Killing charges for the case $b(z) = 0$:

$$S_{BH} = 8\pi\sqrt{\mathcal{T}_0}\left(\sqrt{\bar{\mathcal{S}}_0 + \sqrt{\bar{\mathcal{S}}_0^2 + \frac{\bar{k}}{8}\mathcal{T}_0}} + \sqrt{\bar{\mathcal{S}}_0 - \sqrt{\bar{\mathcal{S}}_0^2 + \frac{\bar{k}}{8}\mathcal{T}_0}}\right), \tag{70}$$

where $\mathcal{T}_0$ and $\bar{\mathcal{S}}_0 = \mathcal{S}_0$ are the charges associated with the Killing vectors $\partial_z$ and $\partial_w$, and

---

[15]We remark that, even though we introduced the construction using three-dimensional backgrounds, it can be generalized and used in any other dimension with minor or no modifications to its philosophy.

$\bar{k} = -2\pi\ell/\kappa_N^2$. This formula thus gives $S_{BH}$ as a function of the zero-mode charges $\mathcal{T}_0$ and $\bar{\mathcal{S}}_0$ and the central extension of the algebra, which is the typical form of a Cardy-like formula. Deriving it from properties of the algebra (66)-(67) would be an extremely interesting result. However, it is worth highlighting an immediate challenge in trying to complete this program. A naive attempt to derive the formula using the subalgebras for which Cardy formulas are known ($\mathfrak{bms}_3$ [3,4,54] and twisted warped conformal [5,55]) fails, because the relevant zero-mode charges are $\mathcal{T}_0$ and $\bar{\mathcal{S}}_0$, which belong to the $\mathfrak{bms}_2$ subalgebra. We do not have a Cardy formula for $\mathfrak{bms}_2$, so obtaining one could also be useful in the present situation. Finally, as already noted at the end of section 4, the algebra (66)-(67) is particularly appealing since it unifies well-known algebras ($\mathfrak{bms}_2$, $\mathfrak{bms}_3$, and twisted warped Witt) into a single structure. This is yet another argument to grant it further study.

More broadly, our construction of asymptotic T-duality through the BTZ / black string example can in principle be generalized to other T-dual pairs in (super)gravity. Given that T-duality can heavily affect the asymptotic structure of a spacetime, and supported by the results obtained for the example developed in the present work, we believe that this can provide a way to obtain novel boundary conditions which lead to interesting symmmetry algebras for a variety of asymptotic behaviors. It is also possible to explore similar ideas in other potentially interesting contexts. One example would be trying to implement an asymptotic notion of T-duality in the language of double field theory [56–58]. Given that this is a particularly well-suited formalism to discuss T-duality equivalent backgrounds, it is conceivable that the asymptotic analysis we have performed in this paper has also an illuminating counterpart in the double field theory language. Another potential avenue would be to implement alternative solution-generating transformations in an asymptotic sense. TsT transformations are probably the first and more natural example to consider, and it would be interesting to explore the effect of TsT transformations applied to the Brown-Henneaux boundary conditions. If one can make sense out of such a construction, the result can also impact our understanding of three-dimensional black strings, given that these can be obtained by TsT transformations of BTZ black holes. Recent works have addressed the question of the effect of such TsT transformations in the asymptotic symmetry algebra using a worldsheet perspective [59], and the results point towards a conservation of the two Virasoro towers. It would be interesting to reproduce such a result from a target space perspective, using methods similar to the ones developed in this work. Finally, it is known that for backgrounds allowing an exact worldsheet description, spacetime and worldsheet symmetries can sometimes be related (this is the case for AdS$_3$ and some of its deformations, and 3d flat space [13, 14, 39, 59–61]). It would be interesting to investigate whether such a construction can be performed for the 3d black string. One aspect to take into account is the fact that the black string metrics are only valid to first order in $\alpha'$, and higher order corrections might have to be taken into account along the lines of [62, 63].

## Acknowledgments

We are grateful to Glenn Barnich, Geoffrey Compère, Dima Fontaine, Oscar Fuentealba, Marc Henneaux, Julio Oliva, Alfredo Pérez, Jakob Salzer and Ricardo Stuardo for insightful discussions and helpful comments on this manuscript.

**Funding information** SD is a Senior Research Associate of the Fonds de la Recherche Scientifique F.R.S.-FNRS (Belgium), is supported in part by IISN - Belgium (convention 4.4503.15) and benefited from the support of the Solvay Family. SD acknowledges support of the Fonds de la Recherche Scientifique F.R.S.-FNRS (Belgium) through the projects PDR/OL C62/5 "Black hole horizons: Away from conformality" (2022-2025) and CDR n°40028632 (2025-2026).

SD is a member of BLU-ULB (Brussels Laboratory of the Universe, blu.ulb.be). JF is partially funded by Beca ANID de Doctorado grant 21212252. AVL acknowledges support from the National Science and Engineering Research Council of Canada (NSERC) and the Simons foundation via a Simons Investigator Award.

## A  Asymptotic symmetries for an exact T-duality

In this appendix, we will prove that the asymptotic symmetry transformations of a phase space with an exact $U(1)$ isometry are preserved under T-duality. We start by writing the general form of a metric, Kalb-Ramond and dilaton fields with a $U(1)$ isometry along the $z$ direction using adapted coordinates

$$\mathrm{d}s^2 = g_{ij}\,\mathrm{d}x^i\,\mathrm{d}x^j + e^{2C}\left(\mathrm{d}z + A_i\mathrm{d}x^i\right)^2, \tag{A.1a}$$

$$B_{ij} = \mathcal{B}_{ij} + B_{[i}A_{j]}, \qquad B_{iz} = B_i, \tag{A.1b}$$

$$\Phi = \phi + \frac{1}{2}C, \tag{A.1c}$$

where the coordinates are split as $x^N = (x^i, z)$. The fields $g_{ij}$, $C$, $A_i$, $B_i$, $\mathcal{B}_{ij}$ and $\phi$ depend only on $x^i$, and the seemingly unnatural definitions of $\phi$ and $\mathcal{B}_{ij}$ simplify later expressions. In particular, Buscher rules in string frame take a simple form. By interchanging

$$A_i \leftrightarrow B_i, \quad \text{and} \quad C \leftrightarrow -C, \tag{A.2}$$

we get a new solution of the equations of motion.

On the other hand, under a symmetry transformation generated by $\xi$ and $\Lambda$ the fields in (A.1) transform as

$$\delta_{\xi,\Lambda}g_{ij} = \mathcal{L}_\xi g_{ij} - A_{(i}g_{j)l}\partial_z\xi^l, \tag{A.3a}$$

$$\delta_{\xi,\Lambda}A_i = \mathcal{L}_\xi A_i + \partial_i\xi^z - A_i\left(\partial_z\xi^z + A_j\partial_z\xi^j\right) + e^{-2C}g_{ij}\partial_z\xi^j, \tag{A.3b}$$

$$\delta_{\xi,\Lambda}C = \mathcal{L}_\xi C + \partial_z\xi^z + A_i\partial_z\xi^i, \tag{A.3c}$$

$$\delta_{\xi,\Lambda}B_i = \mathcal{L}_\xi B_i + B_{ij}\partial_z\xi^j + B_i\partial_z\xi^z + \partial_i\Lambda_z - \partial_z\Lambda_i, \tag{A.3d}$$

$$\delta_{\xi,\Lambda}\mathcal{B}_{ij} = \mathcal{L}_\xi\mathcal{B}_{ij} + B_{[i}\left(\partial_{j]}\xi^z - e^{-2C}g_{j]l}\partial_z\xi^l\right) + B_{[i}A_{j]}\left(\partial_z\xi^z + A_l\partial_z\xi^l\right)$$
$$+ 2\partial_{[i}\Lambda_{j]} - A_{[j}\left(B_{i]}\partial_z\xi^z + B_{i]l}\partial_z\xi^l + \partial_{i]}\Lambda_z - \partial_z\Lambda_{i]}\right), \tag{A.3e}$$

$$\delta_{\xi,\Lambda}\phi = \mathcal{L}_\xi\phi - \frac{1}{2}\left(\partial_z\xi^z + A_i\partial_z\xi^i\right), \tag{A.3f}$$

where we have split the reducibility parameters as $\xi = \xi^i\partial_i + \xi^z\partial_z$, $\Lambda = \Lambda_i\mathrm{d}x^i + \Lambda_z\mathrm{d}z$. Given that we are working on a phase space with a $U(1)$ isometry, we should impose that the parameters $\xi^i$ and $\Lambda$ have not dependence on $z$, and that $\xi^z = \zeta(x^i) + \alpha z$. This simplifies the transformation laws, obtaining

$$\delta_{\xi,\Lambda}g_{ij} = \mathcal{L}_\xi g_{ij}, \tag{A.4a}$$

$$\delta_{\xi,\Lambda}A_i = \mathcal{L}_\xi A_i + \partial_i\zeta - \alpha A_i, \tag{A.4b}$$

$$\delta_{\xi,\Lambda}C = \mathcal{L}_\xi C + \alpha, \tag{A.4c}$$

$$\delta_{\xi,\Lambda}B_i = \mathcal{L}_\xi B_i + \partial_i\Lambda_z + \alpha B_i, \tag{A.4d}$$

$$\delta_{\xi,\Lambda}\mathcal{B}_{ij} = \mathcal{L}_\xi\mathcal{B}_{ij} + 2\partial_{[i}\Lambda_{j]} + B_{[i}\partial_{j]}\zeta + A_{[i}\partial_{j]}\Lambda_z, \tag{A.4e}$$

$$\delta_{\xi,\Lambda}\phi = \mathcal{L}_\xi\phi - \frac{1}{2}\alpha. \tag{A.4f}$$

At this point is easy to see that after applying the transformations (A.2) we get the same transformation laws for each of the fields, up to interchanging $\zeta \leftrightarrow \Lambda_z$ and $\alpha \leftrightarrow -\alpha$ (and recalling the fact the the dilaton has shift invariance).

The solution to the asymptotic symmetry parameters preserving some particular boundary conditions is then the same for one phase space and its T-dual, after the interchange of the gauge parameter and the $z$ component of the diffeomorphism generator. It is worth noticing that this is only proven at the level of the reducibility parameters and not at the level of the charges. Indeed, symmetry transformations with vanishing charge can acquire a charge in the T-dual phase space, as it is shown in section 3.

## B Black string from T-duality: Charges and thermodynamics

In this appendix, we will explicitly show how the three-dimensional black strings fit into the boundary conditions defined by (37). Obtaining the black strings from T-duality was already done in [7], so our task is mainly to transform the results to our current notation. We will also include a brief discussion regarding the thermodynamics of black strings, as it will help clarify the role played by our choice of Cauchy slice when integrating charges.

Let us start from the form of the BTZ metrics. Writing them in a way that fits the boundary conditions (5), and including a free pure gauge term in the $B$-field, we have

$$ds^2 = \frac{\ell^2}{r^2}dr^2 - r^2\left(dx^+ - \frac{\ell^2 L_-}{r^2}dx^-\right)\left(dx^- - \frac{\ell^2 L_+}{r^2}dx^+\right), \tag{B.1a}$$

$$B = \left(\frac{r^2}{C_0} + b(x^a) + \frac{\ell^4}{C_0 r^2}L_+ L_-\right)dx^+ \wedge dx^-, \tag{B.1b}$$

$$\Phi = \frac{1}{4}\log\left(\frac{2}{C_0}\right), \tag{B.1c}$$

where $L_\pm$ are now constants related to the mass and angular momentum of the BTZ black hole by

$$M = \frac{2\pi(L_+ + L_-)}{\kappa_N^2}, \qquad J = \frac{2\pi\ell(L_+ - L_-)}{\kappa_N^2}. \tag{B.2}$$

A direct application of the T-duality Buscher rules along the direction $\xi = \partial_\phi = (\partial_+ - \partial_-)/\ell$ produces the dual field configuration

$$ds^2 = g^4(\hat{r})d\hat{r}^2 + \hat{r}^2 g^2(\hat{r})\left[\left(\frac{2b}{\ell^2} + \frac{4\ell L_+ L_-}{C_0^2 \hat{r}}\right)\left(1 + \frac{b}{2\ell\hat{r}}\right)dz^2 + \frac{\ell}{4\hat{r}}dw^2\right.$$
$$\left. - \left(1 + \frac{b}{\ell\hat{r}} + \frac{\ell^2 L_+ L_-}{C_0^2 \hat{r}^2}\right)dzdw\right], \tag{B.3a}$$

$$B = \frac{\ell^3(L_+ - L_-)}{4C_0^2 \hat{r} g^2(\hat{r})}dz \wedge dw, \tag{B.3b}$$

$$e^{4\Phi} = \frac{\hat{r}_0^2}{\hat{r}^2 g^4(\hat{r})}, \tag{B.3c}$$

in which we have already transformed to the appropriate coordinates for the dual using (38), and $g^2(\hat{r})$ is given by

$$g^2(\hat{r}) = 1 + \frac{\ell}{C_0 \hat{r}}(L_+ + L_-) + \frac{\ell^2}{C_0^2 \hat{r}^2}L_+ L_-. \tag{B.4}$$

By comparing with the boundary conditions (37), it is immediate to read the form of the subleading terms in this solution. Of course, to actually have a solution we need $\xi$ to be an exact Killing vector of the original spacetime, so $b = b(z)$ is a function only of $z$ (directly related to $t$ before the duality).

In order to make contact with the original literature [7, 18], let us use the fact that $b(z)$ can be removed by a diffeomorphism and introduce new coordinates

$$\mathrm{d}z = \mathrm{d}Z, \qquad \mathrm{d}w = \mathrm{d}W + \left(\frac{2b(z)}{\ell^2} - \frac{2(L_+ + L_-)}{C_0}\right)\mathrm{d}Z, \tag{B.5}$$

in terms of which the metric looks like (B.3) but with $b \to \ell^2(L_+ + L_-)/C_0$. This diffeomorphism after T-duality can be regarded as a gauge choice for $b$ in the BTZ solution, (B.1). It turns out this value was the one used in [7], once we take into account that we are in Fefferman-Graham gauge (with the radial piece of the metric fixed to $\ell^2/r^2$). The standard form of the black string appears if we introduce yet another set of coordinates,

$$R = \hat{r}g^2(\hat{r}), \qquad Z = \frac{\sqrt{C_0}(T+X)}{2(L_+L_-)^{1/4}}, \qquad W = \frac{(\sqrt{L_+} + \sqrt{L_-})^2 T + (\sqrt{L_+} - \sqrt{L_-})^2 X}{\sqrt{C_0}(L_+L_-)^{1/4}}, \tag{B.6}$$

in terms of which the solution becomes

$$\mathrm{d}s^2 = \frac{\mathrm{d}R^2}{\left(1 - \frac{\mathcal{M}}{R}\right)\left(1 - \frac{\mathcal{Q}^2}{\mathcal{M}R}\right)} + R^2\left[-\left(1 - \frac{\mathcal{M}}{R}\right)\mathrm{d}T^2 + \left(1 - \frac{\mathcal{Q}^2}{\mathcal{M}R}\right)\mathrm{d}X^2\right], \tag{B.7a}$$

$$B = -\frac{\hat{r}_0^2 \mathcal{Q}}{R}\mathrm{d}T \wedge \mathrm{d}X, \tag{B.7b}$$

$$e^{4\Phi} = \frac{\hat{r}_0^2}{R^2}, \tag{B.7c}$$

where we have introduced new parameters

$$\mathcal{M} = \frac{\ell}{C_0}\left(\sqrt{L_+} + \sqrt{L_-}\right)^2, \qquad \mathcal{Q}^2 = \frac{\ell^2}{C_0^2}(L_+ - L_-)^2. \tag{B.8}$$

Let us analyze the thermodynamics of this black string. This is easy in the $(T, X)$ frame: There is a horizon at $R = \mathcal{M}$ generated by the Killing vector $\partial_T$,[16] under which the Hawking temperature is

$$T_H = \frac{\sqrt{1 - \mathcal{Q}^2/\mathcal{M}^2}}{4\pi} = \frac{(L_+L_-)^{1/4}}{2\pi(\sqrt{L_+} + \sqrt{L_-})}. \tag{B.9}$$

Notice that the asymptotics of this solutions does not allow us to normalize the generator of the horizon in any canonical way at infinity, given that $(\partial_T)^2$ diverges as $R^2$. It is important to remember our choice in order to compare with the results in other frames. In the $(T, X)$ frame, it is natural to compute charges at fixed $T$, in which case we obtain

$$H_T = \frac{\mathcal{M}}{2\kappa_N^2}\Delta X, \qquad H_X = 0, \tag{B.10}$$

for the charges associated to the exact Killing vectors $\partial_T$ and $\partial_X$ ($\Delta X$ is an IR regulator for the integral over $X$, $H_T/\Delta X$ is the physically meaningful quantity giving the energy of the black string per unit length). The first law can now be shown to hold in the form

$$\delta H_T = T_H \delta S_{BH} + \Psi_B \delta Q_B, \tag{B.11}$$

---

[16]We assume $\mathcal{M} \geq |\mathcal{Q}|$ in the following analysis. See the original literature for a discussion of the spacetime structure in other situations [18].

with $S_{BH}$ the Bekenstein-Hawking entropy at constant $T$

$$S_{BH} = \frac{2\pi A_H}{\kappa_N^2} = \frac{2\pi \mathcal{M}}{\kappa_N^2} \sqrt{1 - \frac{\mathcal{Q}^2}{\mathcal{M}^2}}\, \Delta X\,, \tag{B.12}$$

and $\Psi_B$ and $Q_B$ the potential and charge associated to the $B$-field [45],

$$\Psi_B = \frac{\hat{r}_0^2 \mathcal{Q}}{\mathcal{M}}\,, \qquad Q_B = \frac{\mathcal{Q}}{2\kappa_N^2 \hat{r}_0^2} \Delta X\,. \tag{B.13}$$

If we want to think of the black string as embedded in our dual configuration space, satisfying the boundary conditions (37), we should stay in the coordinates (B.3) and work with Cauchy slices at fixed $w$. This is indeed what we did in Section 4. The properties of the horizon are of course independent of the coordinates used, so we still have a horizon at $\hat{r} = \hat{r}_h$ with $\hat{r}_h g^2(\hat{r}_h) = \mathcal{M}$ generated by

$$\xi = \partial_T = \frac{2(L_+ L_-)^{1/4}}{\sqrt{C_0}} \partial_w + \frac{\sqrt{C_0}}{2(L_+ L_-)^{1/4}} \left( \partial_z + \frac{2b(z)}{\ell^2} \partial_w \right). \tag{B.14}$$

Note that $\partial_w$ and the vector within parentheses are the exact Killing vectors of the metric (B.3). The corresponding charges follow from the general covariant phase space formalism described in the main text,

$$H_{(w)} \equiv H[\partial_w] = \frac{\hat{r}_0^2 (L_+ + L_-)}{2\ell \kappa_N^2} \left( \Delta z + \frac{1}{2\hat{r}_0^2 (L_+ + L_-)} \mathcal{B}(\Delta z) \right), \tag{B.15a}$$

$$H_{(z)} \equiv H\left[ \partial_z + \frac{2b(z)}{\ell^2} \partial_w \right] = \frac{8\hat{r}_0^4 L_+ L_-}{\ell^3 \kappa_N^2} \left( \Delta z + \frac{L_+ + L_-}{8\hat{r}_0^2 L_+ L_-} \mathcal{B}(\Delta z) \right), \tag{B.15b}$$

where

$$\mathcal{B}(\Delta z) = \int_{\Delta z} b(z)\, dz\,, \tag{B.16}$$

is the integral of $b(z)$ over a piece $\Delta z$ of the Cauchy slice. In the main text we took $z$ compact to regulate, in which case taking $\Delta z = 2\pi$ so that we integrate over the whole circle, this integral just picks the zero mode of $b(z)$. We assume this is the case from now on and write $\mathcal{B} = 2\pi b_0$. Note also that the non-trivial charges are associated with $T$ and $S$ in the asymptotic Killing vectors (54) (for constant $b$, they are the corresponding zero-mode charges).

We can finally verify the first law in this frame. Of course, the different choice of Cauchy slice did not change the properties of the horizon, in particular the temperature is still

$$T_H = \frac{\sqrt{1 - \mathcal{Q}^2/\mathcal{M}^2}}{4\pi} = \frac{(L_+ L_-)^{1/4}}{2\pi(\sqrt{L_+} + \sqrt{L_-})}\,. \tag{B.17}$$

However, the different slicing does change the entropy, which taking horizon slices of constant $w$ becomes

$$S_{BH} = \frac{16\pi \sqrt{2}\, \hat{r}_0^3 \sqrt{L_+ L_-}\, (\sqrt{L_+} + \sqrt{L_-})}{\ell^2 \kappa_N^2} \left( 1 + \frac{b_0}{4\hat{r}_0^2 \sqrt{L_+ L_-}} \right). \tag{B.18}$$

Similarly, the different slicing affects the $B$-field potential, which is given by $-\xi \cdot B$ pulled-back to a $w$-constant horizon slice now [45],

$$\Psi_B = \frac{\sqrt{2}\, \hat{r}_0^3}{\ell} \frac{(L_+ L_-)^{1/4}(L_+ - L_-)}{(\sqrt{L_+} + \sqrt{L_-})^2} \left( 1 + \frac{1}{4\hat{r}_0^2 \sqrt{L_+ L_-}} \frac{b_0}{2\pi} \right), \tag{B.19}$$

and the three-form charge becomes

$$Q_B = \frac{\mathcal{Q}}{2\kappa_N^2 \hat{r}_0^2} \Delta z = 2\pi \frac{L_+ - L_-}{\ell \kappa_N^2} \,. \tag{B.20}$$

Varying with respect to $L_+$ and $L_-$, one can verify that the first law holds,

$$\frac{2\sqrt{2}\hat{r}_0 (L_+ L_-)^{1/4}}{\ell} \delta H_{(w)} + \frac{\ell}{2\sqrt{2}\hat{r}_0 (L_+ L_-)^{1/4}} \delta H_{(z)} = T_H \,\delta S_{BH} + \Psi_B \,\delta Q_B \,, \tag{B.21}$$

which should not come as a surprise since the first law is a theorem which does not care about how we slice our spacetime.

Let us end with some brief comments and lessons we can extract from this computation for our phase space. All black strings in the form (B.3) are contained within the phase space we built and analyzed in section 4. However, the slicing used there to study the charges is different to the standard slicing that the form (B.7) naturally suggests. This is unavoidable if we want to write a phase space that includes all black strings as they arise from T-duality of BTZ black holes: The map $(\hat{r}, z, w) \to (R, T, X)$ involves the charges $L_\pm$, so we cannot diagonalize the metric in a uniform way for all black strings. It is also illuminating for the discussion to write the entropy (B.18) in terms of the charges,

$$S_{BH} = \frac{2\pi^2 \hat{r}_0^3}{\kappa_N^2 \ell^2} \left( \sqrt{\frac{\left(2q_B^2 + h_{(z)} - 8h_{(w)}^2\right)\left(2q_B^2 + h_{(z)}\right)}{4h_{(w)}^2} + 2h_{(z)}} - \frac{2q_B^2 + h_{(z)} - 8h_{(w)}^2}{2h_{(w)}} \right)$$

$$\times \left( \sqrt{\frac{2q_B^2 + h_{(z)}}{h_{(w)}} + 4q_B} + \sqrt{\frac{2q_B^2 + h_{(z)}}{h_{(w)}} - 4q_B} \right) , \tag{B.22}$$

where we have introduced rescaled charges (per unit length) to simplify the notation,

$$h_{(w)} = \frac{\ell \kappa_N^2}{2\pi \hat{r}_0^2} H_{(w)} \,, \qquad h_{(z)} = \frac{\ell^3 \kappa_N^2}{2\pi \hat{r}_0^4} H_{(z)} \,, \qquad q_B = \frac{\ell \kappa_N^2}{2\pi} Q_B \,. \tag{B.23}$$

If we repeat the construction in the $b_0 = 0$ case, this imposes the relation between the charges $2q_B^2 + h_{(z)} - 8h_{(w)}^2 = 0$, producing a simple-looking formula for the entropy in terms of the Killing charges. To make it even more suggestive, note that for $b_0 = 0$ the charges $\mathcal{S}_0$ and $\bar{\mathcal{S}}_0$ related as in (65) agree, so identifying $\bar{\mathcal{S}}_0 = H_{(w)}$ and $\mathcal{T}_0 = H_{(z)}$ we can write the entropy as

$$S_{BH} = 8\pi \sqrt{\mathcal{T}_0} \left( \sqrt{\mathcal{S}_0 + \sqrt{\bar{\mathcal{S}}_0^2 + \frac{\bar{k}}{8}\mathcal{T}_0}} + \sqrt{\bar{\mathcal{S}}_0 - \sqrt{\bar{\mathcal{S}}_0^2 + \frac{\bar{k}}{8}\mathcal{T}_0}} \right) , \tag{B.24}$$

where $\bar{k} = -2\pi\ell/\kappa_N^2$ characterizes the central extension in the $\{\bar{\mathcal{S}}_m, \mathcal{Q}_n\}$ bracket (alternatively, it is four times the central extension in the $\{\mathcal{S}_m, \mathcal{S}_n\}$ algebra before the redefinition).

## C  Action of the asymptotic Killing vectors on phase space functions

This appendix collects auxiliary results needed to obtain the algebra of charges presented in (63). Brackets are computed according to

$$\{\mathcal{P}_1, \mathcal{P}_2\} = \delta_{\xi_2} \mathcal{P}_1 \,, \tag{C.1}$$

where $\mathcal{P}_1$ and $\mathcal{P}_2$ are any two of the charges in (62), and $\xi_2$ is the asymptotic Killing vector of the form (54) associated with the charge $\mathcal{P}_2$. In essence, brackets are given by the variation under the asymptotic Killing vectors of the charges.

These variations can be deduced from those of the basic phase space functions appearing in the expressions for the charges. The variation of such functions is

$$\delta_\xi A_0 = TA_0' + 2T'A_0 + SA_1 - S', \tag{C.2a}$$

$$\delta_\xi A_1 = TA_1' + T'A_1 + T'', \tag{C.2b}$$

$$\delta_\xi F = T\partial_z F + (S - wT')\partial_w F, \tag{C.2c}$$

$$\delta_\xi \psi = T\partial_z \psi + (S - wT')\partial_w \psi - 2\ell R - 2\ell wQ - \frac{\ell}{4}w^2 T', \tag{C.2d}$$

$$\delta_\xi Z_{zz} = T\partial_z Z_{zz} + 2T'Z_{zz} + (S - wT')\partial_w Z_{zz} + 2(S' - wT'')Z_{zw}$$
$$+ 2\ell\partial_z^2 f - 2\ell\partial_w A\partial_z f + 2\ell\partial_z A\partial_w f - 4\ell A\partial_w A\partial_w f + 2\ell A f, \tag{C.2e}$$

$$\delta_\xi Z_{zw} = T\partial_z Z_{zw} + (S - wT')\partial_w Z_{zw} + \frac{\ell}{4}S' + 2\ell\partial_w A\partial_w f + 2\ell\partial_z\partial_w f - \ell f, \tag{C.2f}$$

where primes denote derivatives with $z$, and $T(z)$, $S(z)$, $R(z)$ and $Q(z)$ are the functions defining the asymptotic Killing vectors (54). To compute the algebra of charges, it is useful to read the following combined variation:

$$\delta_\xi \left( F + \psi - 2Z_{zw} + \frac{\ell}{2}A \right) = T\partial_z \left( F + \psi - 2Z_{zw} + \frac{\ell}{2}A \right) + \ell T'A_0 - 4\ell QA_1 - \ell S' - 4\ell Q', \tag{C.3}$$

where we have simplified the result using equation (61). The reader may also find useful when computing $\{\mathcal{T}_1, \mathcal{T}_2\}$ the following variations under the action of an asymptotic Killing vector for which only $T(z)$ is turned on

$$\delta_T (Z_{zz} + 2AZ_{zw}) = T\partial_z (Z_{zz} + 2AZ_{zw}) + 2T' (Z_{zz} + 2AZ_{zw}) - wT'\partial_w (Z_{zz} + 2AZ_{zw})$$
$$+ \frac{\ell}{2}wT'\partial_z A + \frac{\ell}{2}wT'' \left( A - \frac{w}{2}A_1 \right) + \frac{\ell}{4}w^2 T''', \tag{C.4a}$$

$$\delta_T \left( F + \psi - 2Z_{zw} - \frac{\ell}{2}A_0 \right) = T\partial_z \left( F + \psi - 2Z_{zw} - \frac{\ell}{2}A_0 \right)$$
$$- \ell T' \left( A - \frac{w}{2}A_1 \right) - \frac{1}{2}\ell wT''. \tag{C.4b}$$

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
