# Peer review of "Asymptotic T-duality in three dimensions"

_SciPost Physics, doi:SciPost Phys. 19, 044 (2025)_

## Round 1 · Referee Report · Luca Ciambelli (Referee 2) · 2025-6-12

Report
This paper is well written, and explores an interesting and timely topic.
However, there are some points that should be improved and clarified for the paper to be publishable in SciPost:
1) While it is touched upon throughout the manuscript, the effect on the phase space of the T-duality map in the presence of an isometry remains unclear and mysterious. It is well-explained how such a transformation can be used as a solution-generating technique, but on the other hand it is claimed to be useful at the phase space level (see e.g. top of page 11). This is not so obvious, as it leads to a non-residual symmetry transformation, and thus it maps inequivalent phase spaces. One could (as the authors do) start again in the new solution defining new boundary conditions, but then this is not a procedure, rather an ad hoc -- case by case -- construction. It would be certainly useful to spend more time on this delicate point.
2) Section 2 starts abruptly with a list of equations and quantities that are not sufficiently-well introduced. It is necessary to improve this section, recalling conventions and quantities. Please be more specific about what "the theory" above eq. (1) exactly is, what are the dynamical fields, and the physical intuition.
3) The vocabulary around equations (5a-5c) is confusing. Indeed, these equations are fields falloffs, not boundary conditions. Boundary conditions are something of the form $\delta \alpha\stackrel{S}{=}0$, for a field $\alpha$ and a boundary $S$. Can the authors improve this part?
4) Related to point 1) above, is the vector field (33) the "T-dual" of the original residual symmetries generator? This is not clear, but I believe it is interesting and worth understanding better. A similar question pertains to the vector field (54).
5) From (34a) and (34b), one deduces that T-duality is a physical transformation, generating new non-vanishing charges. However, as we recently explored in 2412.14992, some new charges may well be "kinematical" charges. This means that they have trivial (or unconstrained) fluxes, and/or they come from a corner symplectic potential. This resonates with the discussion around equation (50), in which a new term is added to the symplectic potential, as well as around equations (62a-62d), in which, indeed, the new charges ${\cal R}$ and ${\cal Q}$ have trivial associated fluxes . So these new charges seem to be kinematical, in the language of the afore-mentioned paper. This is further supported by the fact that they do not enter in the entropy formula (70). Then, perhaps, one can make them vanishing using a different boundary term, making T-duality a map that does not generate new charges. This is worth clarifying.
On top of these conceptual questions, some minor typos/comments:
i) Please add references after the sentence "which has played a pivotal role in many developments of AdS/CFT" ii) Below eqs. (29a-29c), the word "together" is perhaps replaceable with the word "semi-directly"? iii) The "well-known fact" mentioned on top of page 10 should come with accompanying references iv) Perhaps a picture at the end of the discussion on page 15 could be clarificatory v) In the paragraph below (62a-62d), importantly, in the sentence "For ${\cal R}$, the implicit and explicit dependences can be shown..." the authors are referring to the charge ${\cal T}$, and not the charge ${\cal R}$. vi) Paragraph before Discussion on page 17: were the authors expecting such a result from T-duality? A bit of physical intuition and contextualization might be useful. vii) Top of page 17 "interesting and new asymptotic symmetry algebras". Aren't those known in 3d? Can the authors explain better what they have in mind related to this point?
Once these comments have been addressed, I will gladly reconsider this paper for publication in SciPost.
Recommendation
Ask for major revision
Report
Recommendation
Publish (meets expectations and criteria for this Journal)

Author: Alejandro Vilar López on 2025-07-06 [id 5624]
(in reply to Report 2 by Luca Ciambelli on 2025-06-12)We would like to start by thanking the referee by his many insightful comments and interesting questions. We will split our reply into the main conceptual points made in the report and the minor typos and comments, and we will use in both cases the numbering introduced by the referee. We will also implement several changes in our manuscript, which will be available in a new resubmission
Main conceptual questions
Minor points and typos
i) We will include more references about the role of the BTZ black hole in AdS/CFT in our resubmission.
ii) The modification will be implemented as suggested.
iii) It is difficult to identify a precise reference because it is a fairly evident fact that can be seen from the Buscher rules (30). We will explain it better in our resubmission so that this is clear.
iv) We will add a figure to help understanding our point.
v) This is indeed a typo which will be appropriately corrected. Thank you for pointing it out!
vi) If anything, we were expecting a result in which the asymptotic symmetry group would not change under T-duality, which is not what happened in the end. This is because T-duality in string theory defines equivalent worldsheet theories, and if the asymptotic symmetry group of the background is expected to capture the symmetry structure of a purportedly dual quantum theory, we would expect it to stay the same. That said, it is true that our theory is far from capturing the full structure of string theory, and the background asymptotics is heavily affected by T-duality, so it is also not completely surprising that we get different results at this level of the analysis. This is why we are not sure about making very clear comments about what to expect or not expect on the basis of physical intuition: we believe exploring the construction in new and different cases can actually help to build a better understanding (which we currently lack) of what is really going on.
vii) We do not think the algebra found in our paper was known, so that is an example of an interesting and new structure. The referee may be referring to the fact that in pure gravity some very general results exist (see 1608.01308 or 1704.07419), but our theory has also matter fields (corresponding to the massless states of the NS-NS sector of a string), and these can significantly modify the analysis. We are not aware of general results in such context. Furthermore, the construction itself can certainly be used beyond three dimensions, so we will add a comment in our resubmission clarifying this.

---

## Round 2 · Referee Report · Luca Ciambelli (Referee 2) · 2025-7-8

Report

The authors addressed my concerns and improved the readability and soundness of the paper. I am glad to recommend this paper for submission in JHEP.

I do not need to revise this manuscript further, but I'd like to point out that my point 3 in the previous report, regarding the semantic of falloffs vs boundary conditions, could be addressed better by the authors. Indeed, the way the authors replied and the discussion around equations 5a-5c can truly lead to confusion. Allow me to explain by providing two different and mutually inconsistent ways of reading these equations as they stand:

1) I might interpret that only the leading term in each expansion is held fixed on the phase space, such that $\delta \eta_{ab}=0$ and $\delta C_0=0$, and the subleading terms are dynamical phase space variable.

2) Conversely, I might interpret these equations as stating that all terms displayed are phase space constants, while the subleading terms not displayed are dynamical: $\delta \eta_{ab}=0=\delta Y_{ab}$, $\delta C_0=0, \delta b=0, \delta \beta=0$, and $\delta \tilde Y=0=\delta \phi$. Note that this is the approach of the original Brown-Henneaux and Strominger papers, which differs from the most recent implicit approach, based more on interpretation 1).

Obviously 1) and 2) are very different phase spaces and, while I perfectly agree with the authors that it is common jargon to confuse falloffs and boundary conditions, it would be better to avoid doing so, in order to make sure that the setup is clear and unambiguously defined.

As I said, this is merely a suggestion aimed at making the manuscript more precise and rigorous, it does not affect the correctness of the paper and its suitability for publication.

Recommendation

Publish (easily meets expectations and criteria for this Journal; among top 50%)

  • validity: -
  • significance: -
  • originality: -
  • clarity: -
  • formatting: -
  • grammar: -

Author:  Alejandro Vilar López  on 2025-07-15  [id 5644]

(in reply to Report 1 by Luca Ciambelli on 2025-07-08)

We thank the referee again for his constructive comments: indeed, we had not realized that as writtten equation (5) could lead to confusions due to some terms being kept fixed while others being dynamical phase space variables. We will add a footnote in a new version to clarify this.

---

## Round 2 · Author Response

In this resubmission, we address several points raised by the referee reports, which we believe have helped to produce an improved version of our manuscript. They are mostly clarifications included in the text.

---

## Round 2 · List of Changes

1. New references have been added at the beginning of the second paragraph of the introduction to justify the pivotal role played by the BTZ black hole in developing different aspects of AdS/CFT.
  2. Around equation (1), both before and after, we have added a more detailed description of what the dynamical fields are and what this theory describes (it is the "low energy string effective theory governing the NS-NS sector").
  3. We have explicitly stated below equation (29) that the algebra has the structure of a semi-direct product.
  4. We have explicitly stated in the paragraph above (34) that one can directly check from Buscher rules that gauge transformations of the B-field become diffeomorphisms after T-duality.
  5. We have added figure 1 (page 16) to help understanding how the sign of A(z,w) affects the asymptotic form of the Cauchy slices.
  6. We have added footnote 12 in page 16 mentioning the recent work on kinematical charges and the interesting possibility that some of the charges we find are of that kind.
  7. We have corrected a typo in the last sentence of the paragraph below (62). It said "For R, ..." when it should be "For T, ...".
  8. We have added two sentences to the first paragraph of the discussion, with the aim to clarify that the construction described in the paper does not directly relate phase spaces, but rather it provides new boundary conditions from existing ones (the analysis of the new phase space has to be done subsequently from scratch).
  9. We have added footnote 14 (page 18) to remark that the construction can be done without significant change in any dimension, even though we illustrated it with a three-dimensional example.

---

## Editorial Decision

published